# Turnover of retroelements and satellite DNA drives centromere reorganization over short evolutionary timescales in *Drosophila*

Cécile Courret[1]*, Lucas W. Hemmer[1], Xiaolu Wei[1], Prachi D. Patel[2], Bryce J. Chabot[2], Nicholas J. Fuda[1], Xuewen Geng[1], Ching-Ho Chang[3], Barbara G. Mellone[2,4], Amanda M. Larracuente[1]*

**1** Department of Biology, University of Rochester, Rochester, New York, United States of America, **2** Department of Molecular and Cell Biology, University of Connecticut, Storrs, Connecticut, United States of America, **3** Division of Basic Sciences, Fred Hutchinson Cancer Center, Seattle, Washington, United States of America, **4** Institute for Systems Genomics, University of Connecticut, Storrs, Connecticut, United States of America

\* ccourret@ur.rochester.edu (CC); alarracu@bio.rochester.edu (AML)

**Data Availability Statement:** All sequences are available from NCBI SRA under Bioproject accession PRJNA1007690. All the BASH pipelines and R scripts used in this study are available on

## Abstract

Centromeres reside in rapidly evolving, repeat-rich genomic regions, despite their essential function in chromosome segregation. Across organisms, centromeres are rich in selfish genetic elements such as transposable elements and satellite DNAs that can bias their transmission through meiosis. However, these elements still need to cooperate at some level and contribute to, or avoid interfering with, centromere function. To gain insight into the balance between conflict and cooperation at centromeric DNA, we take advantage of the close evolutionary relationships within the *Drosophila simulans* clade—*D. simulans*, *D. sechellia*, and *D. mauritiana*—and their relative, *D. melanogaster*. Using chromatin profiling combined with high-resolution fluorescence in situ hybridization on stretched chromatin fibers, we characterize all centromeres across these species. We discovered dramatic centromere reorganization involving recurrent shifts between retroelements and satellite DNAs over short evolutionary timescales. We also reveal the recent origin (<240 Kya) of telocentric chromosomes in *D. sechellia*, where the X and fourth centromeres now sit on telomere-specific retroelements. Finally, the Y chromosome centromeres, which are the only chromosomes that do not experience female meiosis, do not show dynamic cycling between satDNA and TEs. The patterns of rapid centromere turnover in these species are consistent with genetic conflicts in the female germline and have implications for centromeric DNA function and karyotype evolution. Regardless of the evolutionary forces driving this turnover, the rapid reorganization of centromeric sequences over short evolutionary timescales highlights their potential as hotspots for evolutionary innovation.

## Introduction

Cell division is an essential process for the viability of all organisms. Centromeres are chromosomal structures that are indispensable for faithful genome inheritance during cell division—

github: https://github.com/LarracuenteLab/
SimClade_Centromere_2024 and on Dryad
(https://doi.org/10.5061/dryad.1zcrjdg2g). All files
necessary to reproduce the plots are in
supplemental files or on Dryad here: https://doi.
org/10.5061/dryad.1zcrjdg2g.

**Funding:** This work was supported by a National
Science Foundation (https://www.nsf.gov/)
Division of Molecular and Cellular Biosciences
grant MCB-1844693 to AML, with additional
support from the National Institutes of Health
(https://www.nih.gov/) grant R35GM131868 to
BGM. The funders had no role in study design, data
collection and analysis, decision to publish, or
preparation of the manuscript.

**Competing interests:** The authors have declared
that no competing interests exist.

**Abbreviations:** *HTT*, *Het-A*, *TART*, and *TAHRE*;
IDR, irreproducible discovery rate; IF,
immunofluorescence; ORF, open reading frame;
RPM, reads per million; TE, transposable element.

they maintain sister chromatid cohesion and ensure proper chromosome segregation. Centromere defects can lead to the loss of genetic information and are associated with diseases (reviewed in [1]).

In eukaryotes, centromeres are generally marked epigenetically by the presence of the centromere-specific histone H3 variant CENP-A (also known as CID in *Drosophila*) [2–4]. CENP-A plays a central role in centromere identity and function, where it recruits kinetochore proteins, forming a macromolecular structure that allows spindle microtubule attachment [3]. The role of the underlying DNA in centromere function is not well understood, although some sequence properties or the abundance of sequences may contribute to centromere specification and strength (e.g., [5–7]). In most species, centromeres are embedded in repetitive sequences [8], which make it difficult to identify their precise organization. Despite the technical difficulties in studying such complex repetitive structures, recent studies highlight the importance of centromeric DNA in centromere stability and their impact on cell division and disease [9,10].

Centromeres vary widely in size and composition across species, from the point centromeres of *Saccharomyces cerevisiae* to the megabase-sized arrays of the human centromeric α-satellite [8,11]. Although essential for proper chromosome segregation, both CENP-A and centromeric sequences are rapidly evolving, even among closely related species [12–14]. Centromeric DNA is often repetitive and, in general, both higher mutation rates and relaxed selective constraints should lead to rapid evolution [15]. However, this hypothesis assumes that repetitive sequences at centromeres are nonfunctional and the role of centromeric DNA in centromere specificity and function is unclear. That said, the relaxed selection hypothesis cannot explain the rapid evolution and positive selection on centromeric proteins [16], which do have essential functions. One potential explanation for the paradox [12] is that genetic conflicts cause rapid centromere evolution [17]. Stronger centromeres can take advantage of the asymmetry in female meiosis to bias their transmission to the egg, rather than the polar body [18,19]—a process called centromere drive. Centromere proteins, in turn, may evolve rapidly to keep up with rapid DNA sequence evolution at centromeres [16] or restore fair segregation [17]. Centromere drive has been observed in plants [20] and mammals [21–23]. Centromere strength may be partially determined by the ability of centromeric DNA to recruit kinetochore proteins or the spread of CENP-A nucleosomes. For example, some mouse centromeres with larger satellite DNA arrays recruit more centromeric proteins and thus increase their transmission through female meiosis [7]. These satellite repeats thus may behave like "selfish" elements by promoting centromeric chromatin expansion resulting in segregation bias. Centromeric DNA turnover may be driven by the constant replacement of sequences that can acquire more centromere proteins.

Satellite DNAs are not the only type of potentially selfish element occupying centromeres: transposable elements (TEs) are common features of centromeres in some fungi, plants, and animals [24]. TEs can proliferate within and spread between genomes, even when this comes at a cost to their host [25]. While centromere function may not require any specific repeat sequence, some properties of satellite DNAs—e.g., secondary structure [5,6], homogenized arrays, nucleosome-sized repeat units—may facilitate centromere maintenance and function [26]. TEs that insert in centromeres may interrupt otherwise homogenous arrays of satellites and affect centromere function [12,26]. However, the ubiquity of TEs at centromeres across a wide range of taxa suggest that they may instead play a conserved role in centromere specification, or even in centromere function (reviewed in [24,27]), for instance through their active transcription [28]. Therefore, studying centromere evolutionary dynamics over short evolutionary timescales is important for understanding the balance between conflict and cooperation that may exist at centromeric DNA.

The small, but complex genomes of *Drosophila* species make them excellent models for the study of centromere function and evolution. In *Drosophila melanogaster*, centromeres correspond to islands of complex DNA highly enriched in retroelements and flanked by simple tandem satellite repeats [29]. While each centromere has a unique organization, they all share only 1 common component: a non-LTR retroelement called *G2/Jockey-3*. *G2/Jockey-3* is also present in the centromeres of a closely related species, *D. simulans*, suggesting that it could be a conserved feature of *Drosophila* centromeres. While recent reports suggest that *D. melanogaster* and *D. simulans* centromeric regions have distinct satellite repeats [8], we do not know the precise organization of centromeres outside of *D. melanogaster*.

Here, we combine (epi)genomic and cytogenetic approaches to study the evolutionary dynamics of centromeres in 3 closely related species of the *simulans* clade—*D. simulans*, *D. sechellia*, and *D. mauritiana*. These species diverged from each other only ~240,000 years ago and from *D. melanogaster* ~2.4 million years ago (estimated in [30,31]), allowing us to study centromere evolution on 2 different timescales at high resolution. We discover that there has been a complete turnover of centromeric sequences in the ~2.4 Myr since these species diverged from *D. melanogaster*: none of the *D. melanogaster* retroelement-rich centromeres are conserved in the *D. simulans* clade. Instead, 2 complex satellites—a *365-bp* and a *500-bp* tandem satellite repeat—now occupy the centromeres of these species. The centromere-associated *G2/Jockey-3* retroelement remains active in one of the lineages (*D. simulans*) but not the others. We also discover the origins of telocentric chromosomes in *D. sechellia*, where the centromeres of chromosomes *X* and *4* now sit on retroelements with telomere-specific functions. These replacement events imply that centromeres can shift their composition rapidly and between categorically different sequence types: TEs and satellite DNAs. The only chromosomes that do not show these categorical shifts in composition are the Y chromosomes, which have male-specific transmission. This suggests that the selection forces driving rapid centromere evolution are female specific, consistent with recurrent genetic conflicts over transmission through the female germline. Our comparative study of detailed centromere organization has implications for the roles of retroelements and satellites in centromere function and evolution, and karyotype evolution.

## Results

### Satellite emergence at *simulans* clade centromeres

To identify the detailed organization of centromeres in the *simulans* clade, we performed CUT&Tag [32] on embryos from each species (*D. simulans*, *D. sechellia*, and *D. mauritiana*) using a CENP-A antibody. The resulting reads were mapped to versions of each species' genome assembly with improved representation of heterochromatic regions from previous work [33]. Because centromeres sit in highly repetitive genome regions, we analyzed unique and all reads (including multi-mappers) independently (Figs 1 and S1–S3). We identified centromere candidates as the top reproducible CENP-A-enriched contigs (between-replicate irreproducible discovery rate [IDR] < 0.05; S1 Table and S4 Fig). We also used an assembly-free analysis to detect the enrichment of complex repeats in the CENP-A CUT&Tag reads (see Methods). We validated our approach with CUT&Tag in *D. melanogaster*, which recovered the same centromere islands as in Chang and colleagues [29] (S5 Fig).

Like *D. melanogaster*, all 3 *simulans* clade species have a pair of large metacentric "major" autosomes (chromosomes *2* and *3*), a pair of small autosomes (chromosome *4*; referred to as the "dot" chromosome), and a pair of sex chromosomes (*X* and *Y*). For each species, there were 5 contigs that were consistently among the most CENP-A-enriched contigs (S4 Fig), which we considered to be the centromere candidates for each chromosome (S2 Table). We

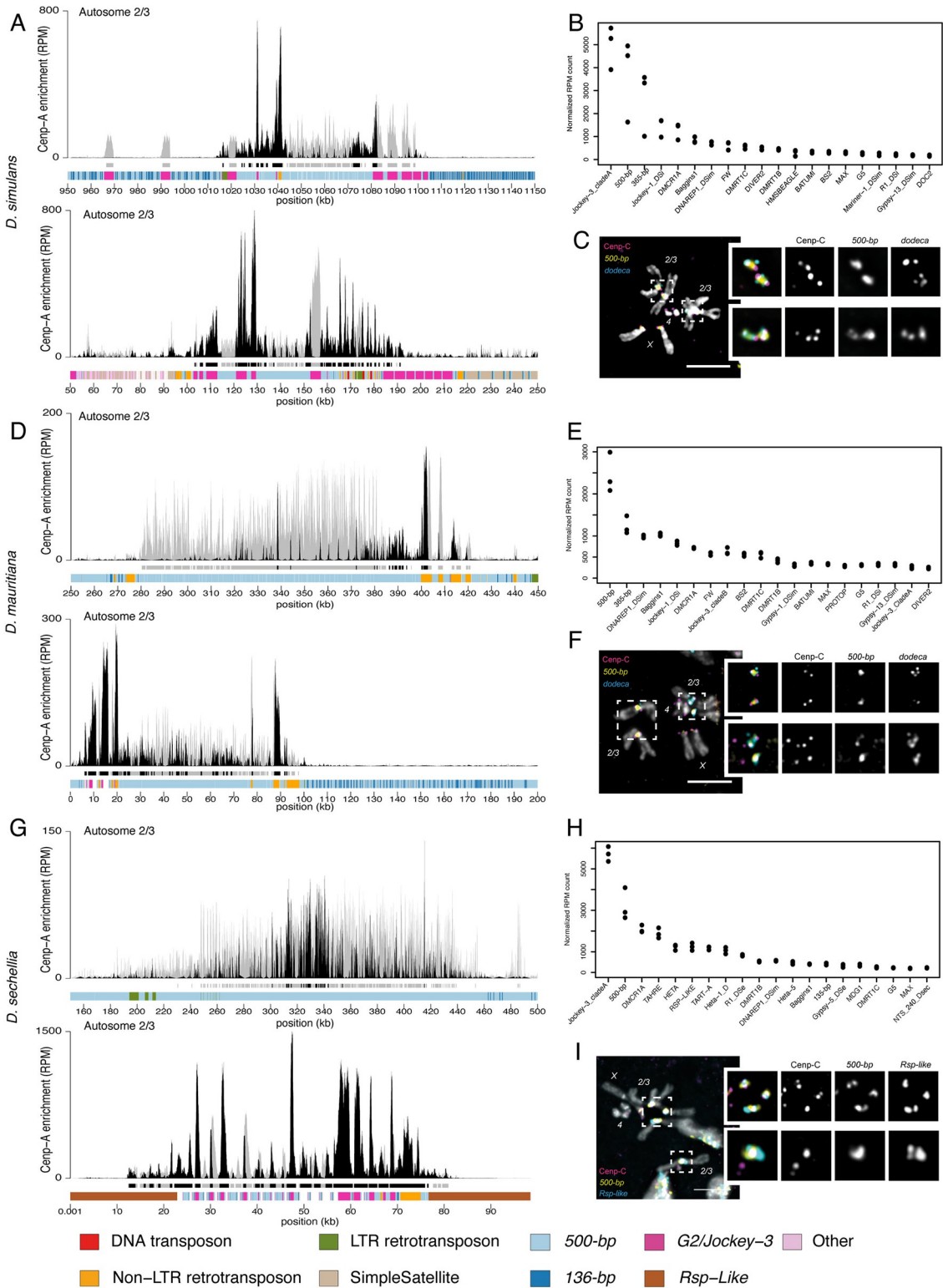

**Fig 1. Centromeres of chromosomes *2* and *3* in *D. simulans*, *D. sechellia*, and *D. mauritiana* are predominantly *500-bp* satellite.**
(A, D, G) CENP-A CUT&Tag enrichment on the centromere candidates for the major autosomes (*2* and *3*) of *D. simulans* (A), *D. mauritiana* (D), and *D. sechellia* (G). The label "Autosome 2/3" indicates that we cannot distinguish between the second and third chromosome centromeres. The y-axis represents normalized CENP-A enrichment in RPM. Black and gray plotted lines represent the enrichment based on uniquely mapping and all reads (including multi-mappers), respectively. The black and gray tracks below each

plot correspond to MACS2 peaks showing significantly enriched regions based on the uniquely mapping and all reads (including multi-mappers), respectively. The precise locations of all peaks are listed in S1 Table. The colored cytoband track at the bottom of the plot shows the repeat organization. The color code is shown in the legend at the bottom of the figure. (B, E, H) Assembly-free analysis showing the normalized enrichment score (in RPM) of CENP-A for complex repeats, including TEs and complex satellites across all centromeres. The Top 20 most enriched repeats are represented for *D. simulans* (B), *D. mauritiana* (E), and *D. sechellia* (H). (C, F, I) IF-FISH on mitotic chromosomes from larval brains with CENP-C antibody and *500-bp* and *dodeca* probes, for *D. simulans* (C) and *D. mauritiana* (F) or *500-bp* and *Rsp-like* probes for *D. sechellia* (I). The insets represent a zoom on each major autosome centromere. Bars represent 5 μm. The data underlying this figure can be found at https://doi.org/10.5061/dryad.1zcrjdg2g [40]. RPM, reads per million; TE, transposable element.

found almost no consistent CENP-A signal outside of these centromere candidates (S4 Fig and S1 Table). None of the *simulans* clade centromere candidates we identified were like *D. melanogaster* centromeres, suggesting a turnover in centromere identity in the ~2.4 My since these species diverged. Instead, both our assembly-based (Figs 1A, 1D, 1G and S1–S3) and assembly-free (Fig 1B, 1E, and 1H) approaches identify the *500-bp* complex satellite among the most CENP-A enriched sequences (Fig 1). The centromere candidate contigs for the major autosomes in *D. simulans*, *D. mauritiana*, and *D. sechellia* (Figs 1A, 1D, 1G, and S1–S3) and the X chromosome in *D. simulans* and *D. mauritiana* (Figs 2 and S1–S3) are mainly comprised of the *500-bp* satellite repeat. This complex satellite was previously identified as being associated with the centromeres in *D. simulans* [8]. While the *500-bp* satellite is the primary repeat type in these *simulans* clade centromeres, they also contain transposable element insertions, including *G2/Jockey-3* (Figs 1A, 1D, 1G, 2, and S1–S3).

The *500-bp* satellite is enriched in, but not specific to, *simulans* clade centromeres, as we also find it in the proximal pericentromeric regions. In *D. melanogaster*, the heterochromatin domain makes up approximately 60 Mb of the genome [34], of which centromeres only represent a small fraction (1/200th [29]). In the *simulans* clade centromeres, the CENP-A domain appears restricted to a 50-kb to 200-kb subset of the *500-bp* satellite array (Figs 1A, 1D, 1G, and 2). This is similar to human centromeres, where the CENP-A domain sits on a subset of *α*-satellite repeats within an array [35]. We also identified a second complex satellite associated with centromere candidates, which we named the *136-bp* satellite. While less abundant, *136-bp* is interleaved with the *500-bp* satellite and is associated with the same centromeres (Figs 1, 2, and S6A).

To validate that the *500-bp* and *136-bp* satellites are associated with the centromeres, we used a cytogenetic approach with IF-FISH on mitotic chromosome spreads from larval brains using Oligopaints targeting each complex satellite [36]. We confirmed the localization of centromeric protein CENP-C, a kinetochore protein that marks the centromeres and has documented overlap with CENP-A [37], on the *500-bp* (Fig 1C, 1F, and 1I) and *136-bp* (S6A Fig) satellites. Because mitotic spreads offer limited resolution, it is challenging to distinguish between the centromeric and proximal pericentromeric domains. However, the *500-bp* signal extends beyond the CENP-C domain, indicating its presence in both the centromeric and proximal pericentromeric regions, consistent with our genome assemblies and CUT&Tag data. While the major autosomal centromeres primarily consist of the same complex satellites in the 3 species, the distal pericentromere appears more divergent. In *D. simulans* and *D. mauritiana*, the major autosomal pericentromeres contain the *dodeca* satellite (Fig 1C and 1F), while in *D. sechellia* they contain the *Rsp-like* satellite (Fig 1I). We also found the *Rsp-like* satellite on the X pericentromere of *D. simulans* (Fig 2A), which was absent in *D. mauritiana* (Fig 2B) [38,39]. The combination of satellites flanking the CENP-A domain (Figs 1C, 1F, 1I, and 2) allows us to assign the *500-bp* enriched contigs to either the major autosomes (Fig 1A, 1D, and 1G) or the X chromosome (Fig 2). Unfortunately, we cannot morphologically distinguish between the chromosomes *2* and *3* because of their similarity.

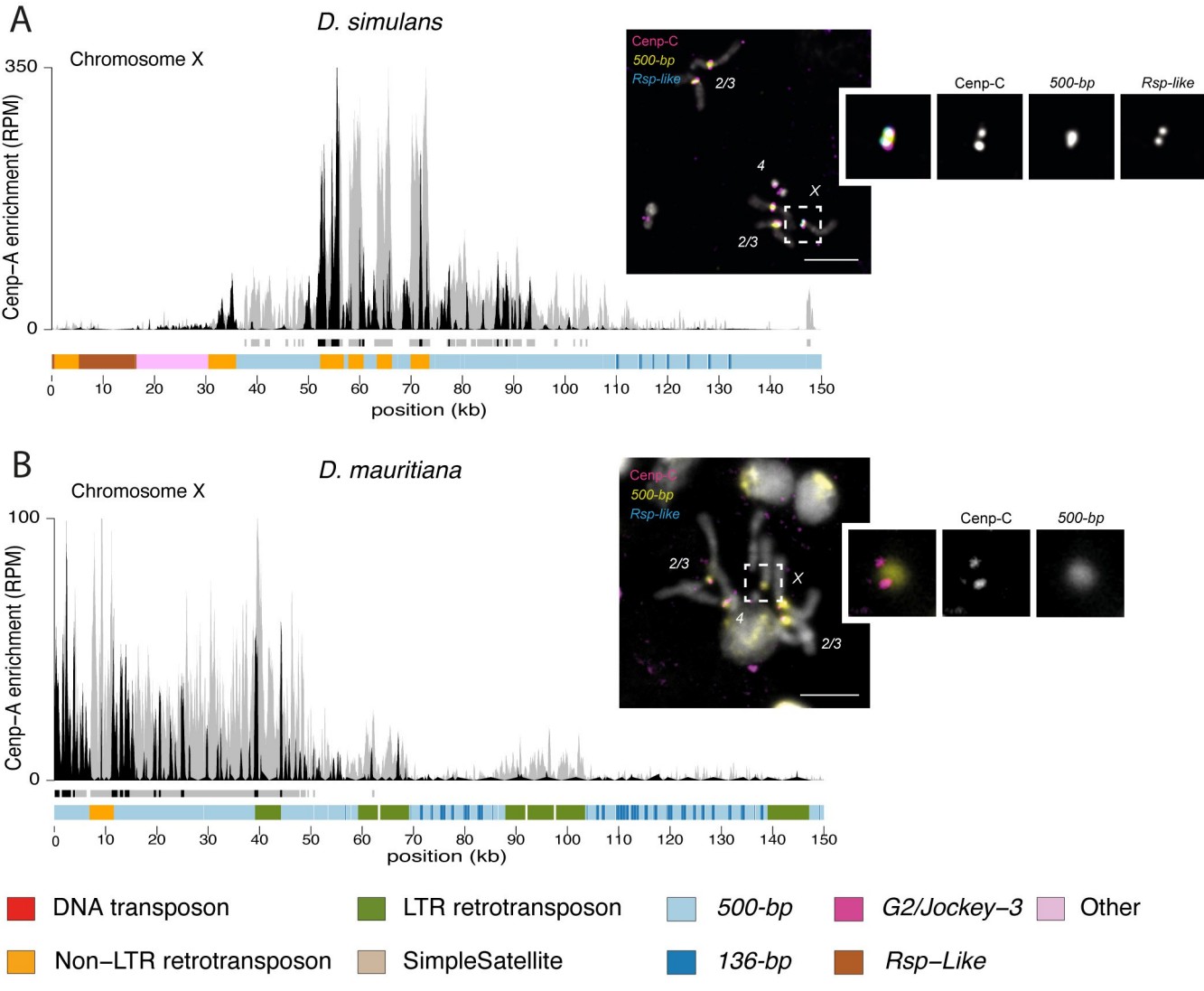

**Fig 2. X chromosome centromeres in *D. simulans* and *D. mauritiana* are enriched in *500-bp* satellite.** The left panel shows the CENP-A CUT&Tag enrichment on the X centromere candidate in *D. simulans* (A) and *D. mauritiana* (B). The y-axis represents the normalized CENP-A enrichment in RPM. Black and gray plotted lines represent the enrichment based on uniquely mapping and all reads (including multi-mappers), respectively. The black and gray tracks below each plot correspond to MACS2 peaks showing significantly enriched regions based on the uniquely mapping and all reads (including multi-mappers), respectively. The precise locations of all peaks are listed in S1 Table. The colored cytoband at the bottom of the plot shows the repeat organization. The color code is shown in the legend at the bottom of the figure. The right panel shows IF-FISH on mitotic chromosomes from larval brains with CENP-C antibody and *500-bp* and *Rsp-like* probes. The inset represents a zoom on each X chromosome centromere. Bars represent 5 μm. The data underlying this figure can be found at https://doi.org/10.5061/dryad.1zcrjdg2g [40]. RPM, reads per million.

We used a BLAST approach to explore origins of the *500-bp* and *136-bp* centromeric complex satellites and did not find any evidence of their presence outside of the *D. simulans* clade, even as single copy sequences (S3 and S4 Tables). For example, in *D. melanogaster*, the best hit had 85% identity with the *500-bp* consensus sequence but only covered 106 bp of the query (S3 Table). This suggests that these satellites emerged after the divergence between *D. melanogaster* and the *D. simulans* clade 2.4 Mya [30,31], although it is possible that the primary sequence emerged earlier but was lost in *D. melanogaster*. In either case, these satellites recently expanded in the *D. simulans* clade centromeres (S7 Fig).

## Dot chromosome centromeres are enriched with a chromosome-specific complex satellite

In *D. simulans* and *D. mauritiana*, the centromere of the small autosomal dot chromosome (i.e., Chromosome 4) contains a different complex satellite: the *365-bp* satellite (Fig 3). The *365-bp* satellite shares no homology with the *500-bp* satellite, suggesting an independent origin. This repeat is consistently enriched in CENP-A chromatin in both our assembly-based (Fig 3) and assembly-free (Fig 1B and 1E) approaches. The CENP-A domain is restricted to the *365-bp* satellite and flanked by the *AATAT* satellite on at least 1 side (Fig 3), which is confirmed by our FISH with CENP-C IF on chromosome spreads (Fig 3 insets). Unlike the *500-bp*

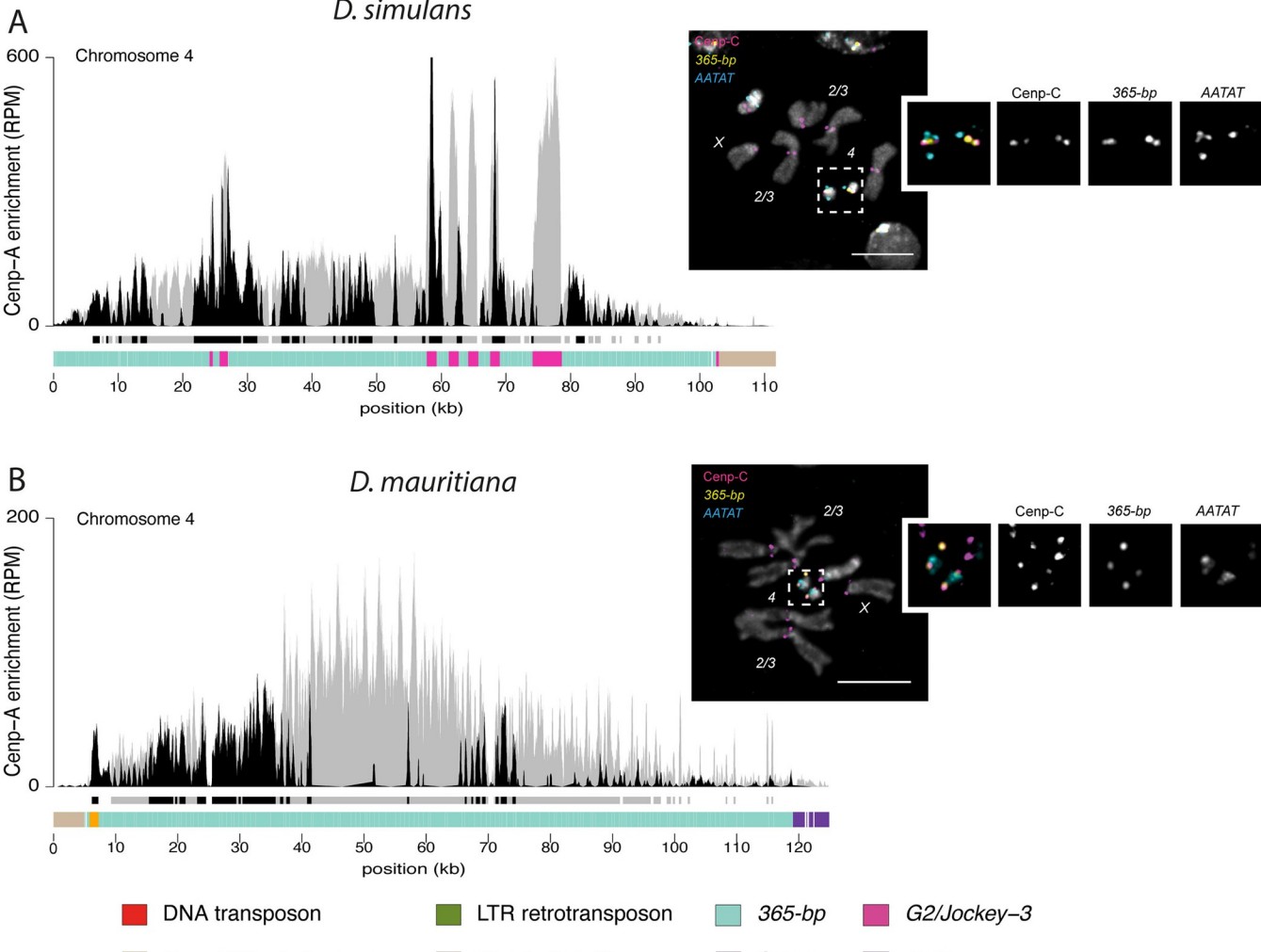

**Fig 3. Dot chromosome centromeres in *D. simulans* and *D. mauritiana* are enriched in *365-bp* satellite.** The left panel represents the CENP-A CUT&Tag enrichment in *D. simulans* (A) and *D. mauritiana* (B). The y-axis represents the normalized CENP-A enrichment in RPM. Black and gray plotted lines represent the enrichment based on uniquely and multi-mapping reads, respectively. Black and gray plotted lines represent the enrichment based on uniquely mapping and all reads (including multi-mappers), respectively. The black and gray tracks below each plot correspond to MACS2 peaks showing significantly enriched regions based on the uniquely mapping and all reads (including multi-mappers), respectively. The precise locations of all peaks are listed in S1 Table. The colored cytoband track at the bottom of the plot shows the repeat organization. The color code is shown in the legend at the bottom of the Fig. The right panel represents the IF-FISH on mitotic chromosomes from the larval brain with CENP-C antibody and *365-bp* and *AATAT* probes. The insets represent a zoom on each dot chromosome centromere. Bars represent 5 μm. The data underlying this figure can be found at https://doi.org/10.5061/dryad.1zcrjdg2g [40]. *HTT*, *Het-A*, *TART*, and *TAHRE*; RPM, reads per million.

satellite, *365-bp* is specific to the dot chromosome centromere. We do not find evidence of the *365-bp* satellite outside of one CENP-A enriched contig in each assembly (Fig 3), consistent with the FISH signals (Fig 3 insets).

We used a BLAST-based approach to explore the origin of the *365-bp* satellite and did not find evidence of this satellite outside of the *D. simulans* clade species (S5 Table). For example, in *D. melanogaster*, the best hit had 82% identity with the *365-bp* consensus sequence but was only 57 bp long (S5 Table) suggesting that, like the *500-bp* satellite, the *365-bp* satellite emerged after the split with *D. melanogaster* and likely emerged at the dot centromeres in the ancestor of the *D. simulans* clade (S7 Fig). One intriguing possibility is that *365-bp* may share origins with (or be derived from) a sequence similar to those currently at *D. melanogaster* centromeres, as some short sequence fragments with similarity to a subset of the *365-bp* satellite are on *D. melanogaster* X and dot centromeres (S5 Table).

Interestingly, *365-bp* was lost from *D. sechellia*: we did not find cytological (S6B Fig) or genomic evidence of this satellite, even as a single copy sequence in the genome assembly, the genomic Illumina reads (S5 Table), or the CENP-A CUT&Tag reads (Fig 1H). However, the pericentromeric *AATAT* satellite appears to be conserved (S6B Fig).

## Centromere shifts to telomere-specialized retroelements: Telocentric chromosomes in *D. sechellia*

In *D. sechellia*, the dot and X chromosome are distinct from those of *D. simulans* and *D. mauritiana*. We did not identify any *500-bp*-enriched contig that might correspond to the X chromosome centromere and *365-bp* is completely missing from the *D. sechellia* genome.

Instead, we identified 2 *D. sechellia* contigs that are significantly enriched for CENP-A containing an array of non-LTR retroelements well known for their role at telomeres: *Het-A*, *TART*, and *TAHRE* (also known as the *HTT* elements) [41]. The *HTT* elements are also among the most CENP-A-enriched elements in our assembly-free approach (Fig 1H). *Drosophila* species lack telomerases, instead, telomere size and integrity are maintained by the transposition activity of *HTT* retroelements [41]. *HTT* elements have specialized functions at telomeres of many *Drosophila* species, including *D. melanogaster* and the *D. simulans* clade [41].

On one *HTT*-CENP-A enriched contig, the *HTT* domain is adjacent to the *500-bp* satellite, suggesting that it corresponds to the X chromosome centromere (Fig 4C). However, in *D. sechellia*, CENP-A is enriched on the *HTT* domain instead of the *500-bp* satellite (Fig 4A), suggesting a repositioning of the centromere to the retroelements that normally occupy the telomere. Similarly, on the second *HTT*-CENP-A enriched contig, the CENP-A domain is flanked by a simple ATAG satellite only found on X and dot chromosomes [42] (Fig 4B). Thus, we infer that this second contig corresponds to the dot chromosome centromere.

To validate our observations, we designed Oligopaints targeting the *HTT* array on the X and dot chromosome centromere candidates in *D. sechellia*. The IF-FISH on mitotic chromosomes from larval brains confirmed that the centromeric protein CENP-C is indeed associated with the *HTT* domain on both the X and dot chromosomes. The *500-bp* satellite appears adjacent to the *HTT* on the X chromosome (Fig 4C).

To visualize these regions at higher resolution, we performed IF-FISH on stretched chromatin fibers using a CENP-A antibody and Oligopaints targeting the *500-bp* satellite and the *HTT* elements. These fibers confirm that CENP-A nucleosomes are seated on the *HTT* domain, and are flanked by, but do not overlap, the *500-bp* satellite (Fig 4D). On average, 89.82 ± 19.4% of the CENP-A signal overlaps with the *HTT* signal, while only 6.2 ± 13.6% overlaps with the *500-bp* signal (S6 Table). The chromatin fibers appear to end shortly after the

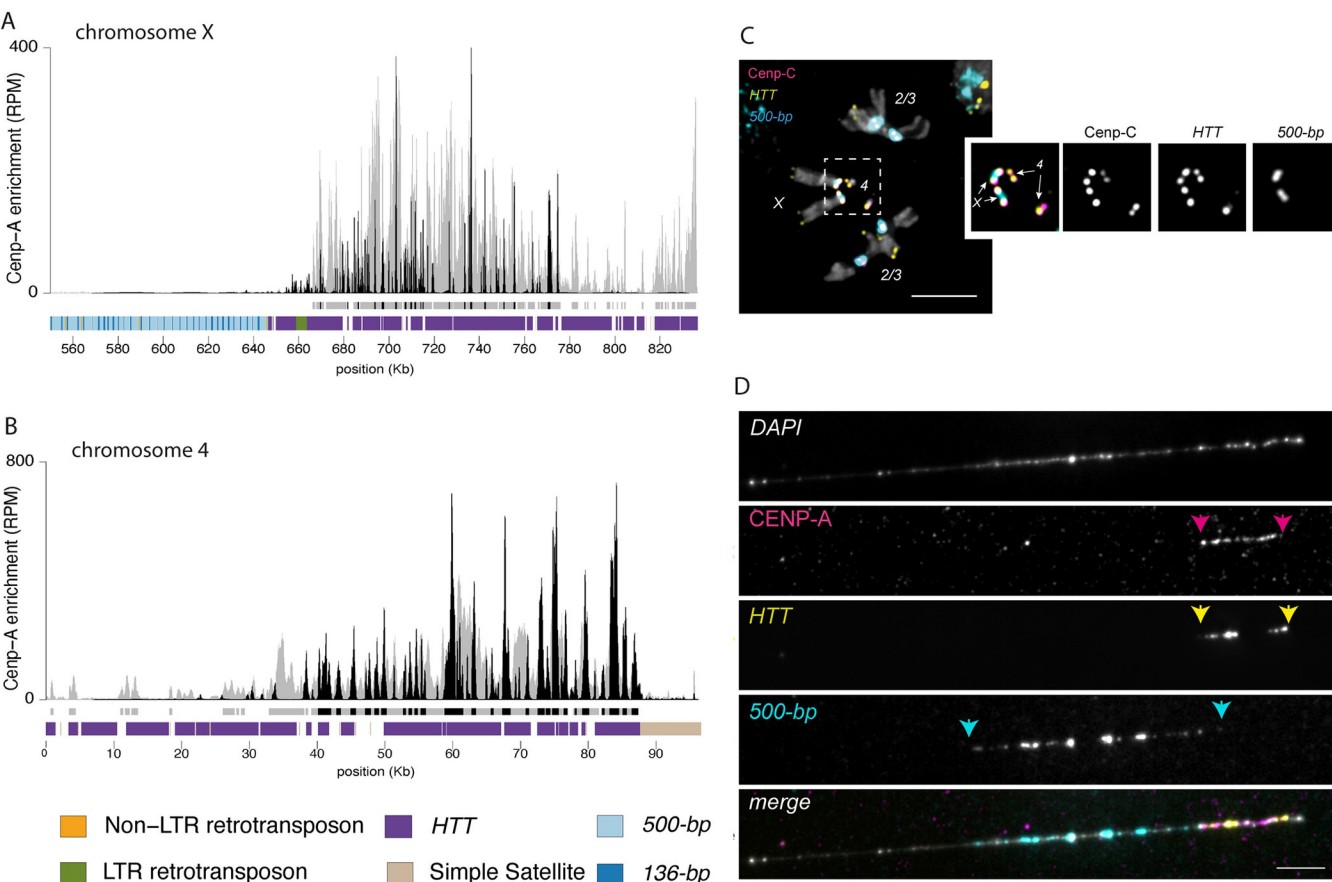

**Fig 4. The Dot and X chromosome centromere in *D. sechellia* are telocentric.** CENP-A CUT&Tag enrichment along the X (**A**) and dot (**B**) chromosome centromeres. The y-axis represents the normalized CENP-A enrichment in RPM. Black and gray plotted lines represent the enrichment based on uniquely and multi-mapping reads, respectively. Black and gray plotted lines represent the enrichment based on uniquely mapping and all reads (including multi-mappers), respectively. The black and gray tracks below each plot correspond to MACS2 peaks showing significantly enriched regions based on the uniquely mapping and all reads (including multi-mappers), respectively. The precise locations of all peaks are listed in S1 Table. The colored cytoband track at the bottom of the plot shows the repeat organization. The color code is shown in the legend at the bottom of the figure. (**C**) IF-FISH on mitotic chromosomes from the larval brain with CENP-C antibody and *500-bp* and *HTT* probes. The inset represents a zoom on the X and dot chromosome centromeres. Bar represents 5 μm. (**D**) IF-FISH on chromatin fibers from the larval brain with CENP-A antibody and *500-bp* and *HTT* probes, representing the telocentric X chromosome of *D. sechellia*. Bar represents 5 μm. The data underlying this figure can be found at https://doi.org/10.5061/dryad.1zcrjdg2g [40]. *HTT*, *Het-A*, *TART*, and *TAHRE*; RPM, reads per million.

CENP-A/*HTT* signal, strongly suggesting that the centromere is on a telomeric *HTT* array, making these chromosomes telocentric (Fig 4D). In some fibers, we observed a lack of CENP-A/*HTT* signal at the very ends, similar to what we show in Fig 4D. It is possible that there is a small amount of non-*HTT* sequence distal to the *HTT* signal on these chromatin fibers. However, we believe that the absence of *HTT* signal at the fiber ends is likely a technical artifact due to the loss of the FISH signal, as this observation was variable across fibers (see S8 Fig). Regardless, the overlap between CENP-A and *HTT* signal confirms that these centromeres are telocentric.

We also observed patterns from stretched chromatin fibers consistent with our predictions for the other chromosome centromeres (S8 Fig). On the dot chromosome 73.02 ± 32.76% of the CENP-A signal overlaps with the *HTT* signal, with no *500-b*p signal nearby (S8 Fig and S6 Table). On the autosomes, 100% of the CENP-A signal overlaps with the *500-b*p signal (S8 Fig and S6 Table).

Interestingly, the dot chromosome centromere of *D. mauritiana* is flanked by the *AATAT* satellite on one side and by the *HTT* on the other side (Fig 3B). Unfortunately, the contig is not long enough to establish how long the *HTT* domain continues after the centromere, but it suggests that in *D. mauritiana*, and possibly *D. simulans*, both centromeric and telomeric domains are very close to each other.

It was very surprising to find the centromeric protein associated with telomeric sequences, as centromeres and telomeres are chromosome domains with distinct functions. Although both the X and the dot chromosomes were considered to be acrocentric chromosomes based on the similarity in karyotype with *D. melanogaster* [43,44], our high-resolution approach allowed us to reveal that these chromosomes are actually telocentric. We demonstrate here that centromeres can share sequence components with telomeres [45]. Currently, we lack the ability to ascertain whether the centromere and telomere share a common domain or exist as separate domains within the *HTT* array.

## The Y chromosome centromeres are unusual

In all 3 species analyzed, the Y chromosome centromeres are unique in their composition and organization compared to the rest of the centromeres in the genome. Unlike the other chromosomes, we did not identify any complex satellites associated with the Y chromosome centromere. Instead, CENP-A is enriched in a region with high density of transposable elements (Fig 5). Despite being mainly enriched in retroelements, the Y chromosomes from each species have a unique composition (Fig 5 and S7 Table). For example, the most abundant elements associated with the Y centromere are *HMSBEAGLE* and *Jockey-1* in *D. simulans*, *mdg4* in *D. mauritiana*, and *R1* and *G2/Jockey-3* in *D. sechellia* (S7 Table). Interestingly, centromeric sequences form higher order repeats in both *D. simulans* and *D. sechellia*, but not in *D. mauritiana* (S9 Fig).

To validate our candidate Y centromeres, we designed Oligopaints specific to the Y contig of each species (*cenY*). We performed IF-FISH on mitotic chromosomes with a CENP-C antibody and the Oligopaint targeting the putative Y centromeres. Our Oligopaints give a signal specific to a unique region of the Y chromosome which consistently co-localizes with the CENP-C signal (Fig 5), confirming the Y chromosome centromeres.

While simple satellites are present within the pericentromeric region of all the other chromosomes, we do not find any simple satellites in the flanking region of the Y centromere (Fig 5). This is surprising, especially given that these Y chromosomes in these species are highly enriched in simple satellites in general [46,47].

## *G2/Jockey-3* is associated with centromeres within the *D. simulans* clade

In *D. melanogaster*, the only common sequence among all centromeres is *G2/Jockey-3* [29]. We asked if this element was also found within the *simulans* clade centromeres. In *D. simulans*, *G2/Jockey-3* is the most enriched repeat among the CENP-A reads (Fig 1B). We identified *G2/Jockey-3* insertions in each centromere except for the X chromosome, where it directly flanks the centromere (Fig 2A). We confirmed the presence of *G2/Jockey-3* at each centromere by IF-FISH on mitotic chromosomes (Fig 6C). In *D. sechellia*, *G2/Jockey-3* is also the most enriched repeat in CENP-A chromatin (Fig 1H); however, we only detect it on the Y chromosome and one of the autosomal centromeres (Figs 1G, 5C, and 6C). Similarly, in *D. mauritiana*, *G2/Jockey-3* is associated with only one of the autosomal centromeres (Figs 1D and 6C) and is less enriched than in the two other species (Fig 1E). This suggests that the association of *G2/Jockey-3* with the centromere was lost.

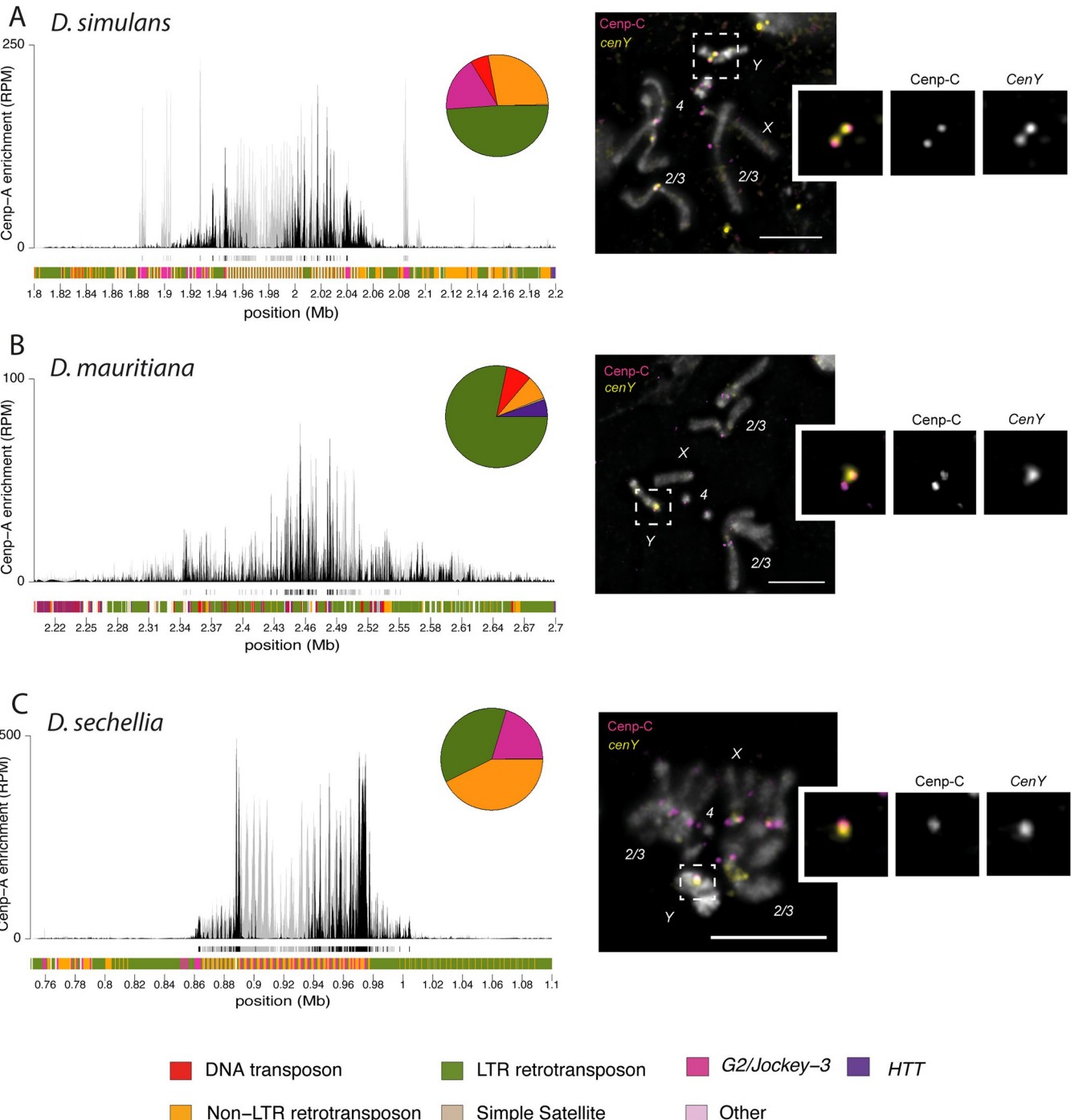

**Fig 5. The Y chromosome centromeres of *D. simulans*, *D. mauritiana*, and *D. sechellia* are rich in transposable elements.** The left panel shows the CENP-A CUT&Tag enrichment for the Y centromere of *D. simulans* (A), *D.mauritiana* (B), and *D. sechellia* (C). The y-axis represents the normalized CENP-A enrichment in RPM. Black and gray plotted lines represent the enrichment based on uniquely mapping and all reads (including multi-mappers), respectively. The black and gray tracks below each plot correspond to MACS2 peaks showing significantly enriched regions based on the uniquely mapping and all reads (including multi-mappers), respectively. The precise locations of all peaks are listed in S1 Table. The colored cytoband track at the bottom of the plot shows the repeat organization. The pie chart on the top represents the repeat composition of the CENP-A domain. The color code of the cytoband and pie chart is shown in the legend at the bottom of the figure. The right panel shows the IF-FISH on mitotic chromosomes from the larval brain with CENP-C antibody and *cenY* Oligopaints specific to each species' centromere. The insets represent a zoom on each Y chromosome centromere. Bar represents 5 μm. The data underlying this figure can be found at https://doi.org/10.5061/dryad.1zcrjdg2g [40]. *HTT*, *Het-A*, *TART*, and *TAHRE*; RPM, reads per million.

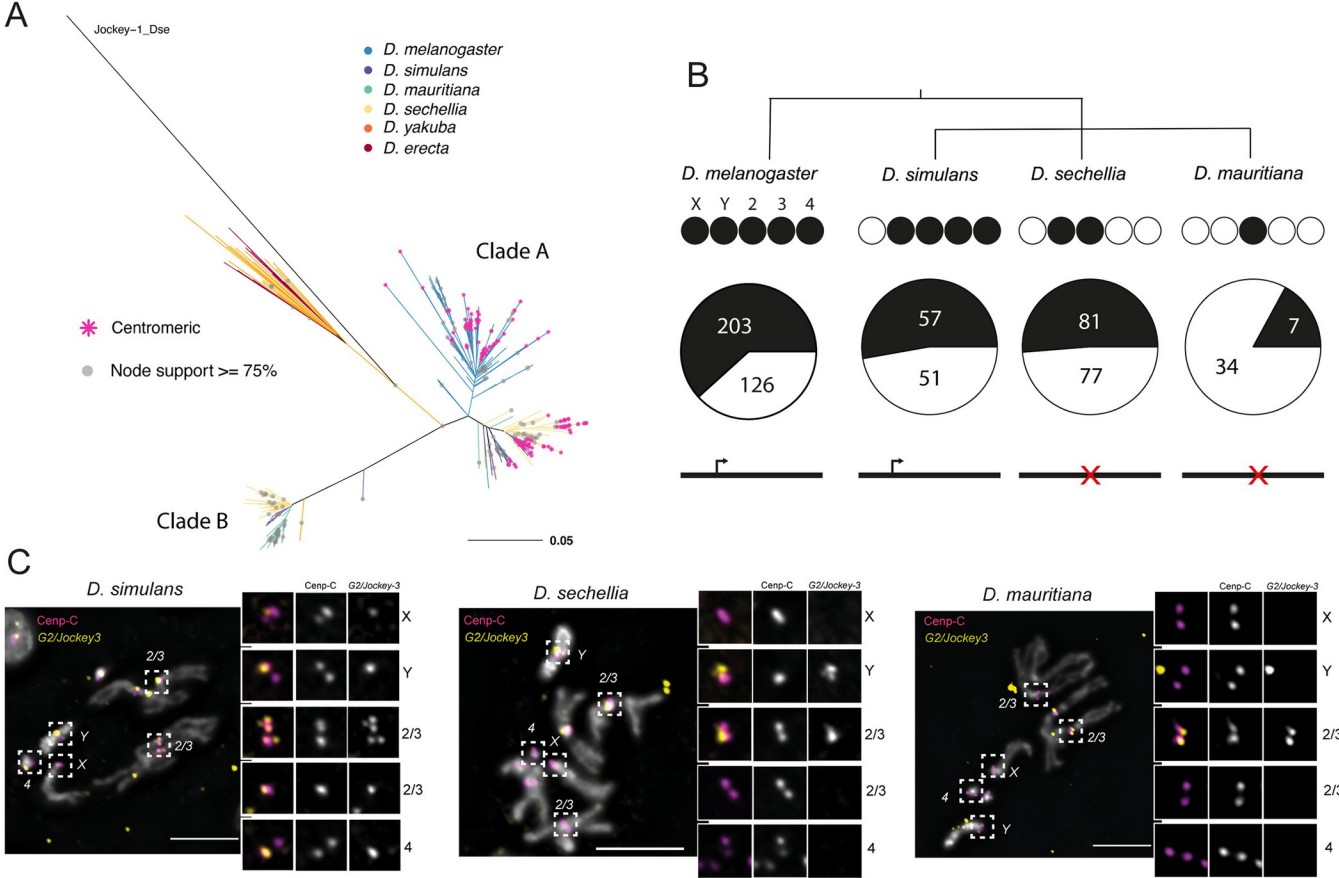

**Fig 6. *G2/Jockey-3* is associated with the centromeres within the *D. simulans* clade.** (**A**) Maximum likelihood phylogenetic tree of *G2/Jockey-3* ORF2 from *D. melanogaster*, *D. simulans*, *D. sechellia*, *D. mauritiana*, *D. yakuba*, and *D. erecta*. *G2/Jockey-3* within the *simulans* clade species diverged into 2 different clades, one that is more closely related to the *D. melanogaster* elements (clade "A") and one that is more divergent (clade "B"). Centromeric insertions are indicated by a pink * at the tip of the branch. We do not know centromere identity in *D. yakuba* and *D. erecta*. (**B**) ORF2 conservation analyses of the clade "A" *G2/Jockey-3* centromere-associated clade. The circles below the species name represent each centromere. Centromeres containing *G2/Jockey-3* insertions (based on CUT&Tag and FISH) are shown in black. The pie chart represents the proportion of centromeric (black) and non-centromeric (white) insertions among the clade "A" *G2/Jockey-3* within each species' genome, where we indicate the number of insertions within the pie charts. The consensus sequence of *G2/Jockey-3* ORFs is schematized below the pie chart, indicating that only *D. melanogaster* and *D. simulans* consensus sequences have an intact ORF2. (**C**) IF-FISH on mitotic chromosomes from the larval brain with CENP-C antibody and *G2/Jockey-3* probes showing consistent centromere-association in *D. simulans*, but not in *D. mauritiana* and *D. sechellia*. In *D. simulans*, the *G2/Jockey-3* insertions on the X chromosome are adjacent to the CENP-A domain, rather than within. The inset represents a zoom on each centromere. Bars represent 5μm. The data underlying this figure can be found at https://doi.org/10.5061/dryad.1zcrjdg2g [40]. ORF, open reading frame.

To better understand the evolutionary history of this specific retroelement, we inferred the phylogeny for all *G2/Jockey-3* ORFs in the *D. melanogaster* clade assemblies. *G2/Jockey-3* has 2 open reading frames (ORFs), but we only used ORF2 for inferring phylogenies, as ORF1 is more evolutionarily labile across species [48]. While all *D. melanogaster G2/Jockey-3* insertions cluster together in a unique clade, the *D. simulans* clade insertions separate into 2 different clades, which we designate as clade "A"—with sequences more closely related to *D. melanogaster G2/Jockey-3*—and clade "B" (Figs 6A and S10). Within each clade, insertions largely form species-specific clusters. All centromeric insertions are part of the clade "A" and retain a conserved ORF2. Like *D. melanogaster*, clade "A" *G2/Jockey-3* insertions are enriched at centromeres (Fig 6B). That is, 53% of clade "A" *G2/Jockey-3* insertions are centromeric in *D. simulans* and *D. sechellia*, which is more than expected if these TEs were randomly distributed in the genome (Fisher's exact tests: $P_{sim} < 10^{-16}$; $P_{sec} < 10^{-16}$). The enrichment is less

pronounced in *D. mauritiana* (17%; $P_{mau}$ = 0.0567). However, the consensus ORF is incomplete in *D. sechellia* and *D. mauritiana*, implying that most clade "A" *G2/Jockey-3* copies are degenerated in these species, in line with their inconsistent association with centromeres. These findings suggest that a subset of *G2/Jockey-3* elements likely had centromere-biased insertion activity in the *D. melanogaster* clade ancestor. This activity may have continued after the speciation event between *D. melanogaster* and the *D. simulans* clade but was lost in *D. sechellia* and *D. mauritiana* lineages, explaining the inability of *G2/Jockey-3* to jump into centromeres. While the clade "B" appears to have been recently active in the *simulans* clade, none of the insertions are centromeric. This clade was either lost from *D. melanogaster* or may have been introduced into the *D. simulans* ancestor through a horizontal transfer event. The latter appears to be more likely as we find fragmented copies of *G2/Jockey-3* from *D. yakuba* that cluster with clade "B." However, we do not have sufficient node support to draw strong conclusions about the origins of this clade. Taken together, our data suggest that the clade "A" *G2/Jockey-3* targeted the centromeres for insertion in both *D. melanogaster* and the *D. simulans* clade specie despite having distinct centromeric sequences, suggesting that this element may preferentially target centromeric chromatin rather than particular DNA sequences.

## Discussion

In the last decade, several studies have shed light on the rapid evolution of centromere sequences in a wide range of species [11]. Centromeres are dynamic in their genomic location and can rapidly diverge in sequence between related species. However, they generally consist of different variants of the same type of repeat element (either retroelements or satellites) [49–56] therefore maintaining a certain homogeneity among closely related species. For example, the centromeres of human and its closely related species—chimpanzee, orangutan, and macaque—are populated by different subfamilies of the $\alpha$-satellite repeat [51,52]. *Arabidopsis* species, *A. thaliana* and *A. lyrata*, also experienced a turnover of centromere sequences since their divergence, but between related satellites [57]. In this study, we reveal that *Drosophila* centromeres appear to experience recurrent turnover between different repeat types over short evolutionary timescales (Fig 7). We hypothesize that the ancestral centromeres resembled the retroelement-rich islands of *D. melanogaster* and that centromere turnover in the *D. simulans* clade species was facilitated by the rapid spread of the *500-bp* and *365-bp* complex satellite repeats (<2.4 Mya). The only retroelement countering the domination of these complex satellites and preventing the complete homogenization of centromeres is *G2/Jockey-3*. Following the emergence of the centromeric complex satellites, the centromere shifted to the neighboring telomeric *HTT* in *D. sechellia* on the X and dot chromosomes (in <240 Kya). This rapid evolution of centromere sequences seems to be a general feature of the *Drosophila* genus [58]. One clade where centromere evolution seems to experience similar dynamics is in the Equus genus, where evolutionarily new centromeres appear in chromosomal regions free from satellite DNAs (e.g., [59]).

The dramatic shifts in centromere composition that we described here raise questions about the role of DNA sequences in centromere function and the dynamic processes driving such shifts. There are 2 primary hypotheses that could explain such rapid centromere turnover: (1) relaxed selective constraints on centromeric DNA; and (2) positive selection—either for particular DNA sequences that make "better" centromeres or due to selfish DNA sequences trigger evolutionary arms races. It is possible that the rapid turnover of centromeric sequences is due to neutral processes, as satellite DNAs are known to rapidly expand and contract through recombination-mediated processes (reviewed in [15]). Transposable elements are generally regarded as deleterious, and therefore have the potential to create conflict in the

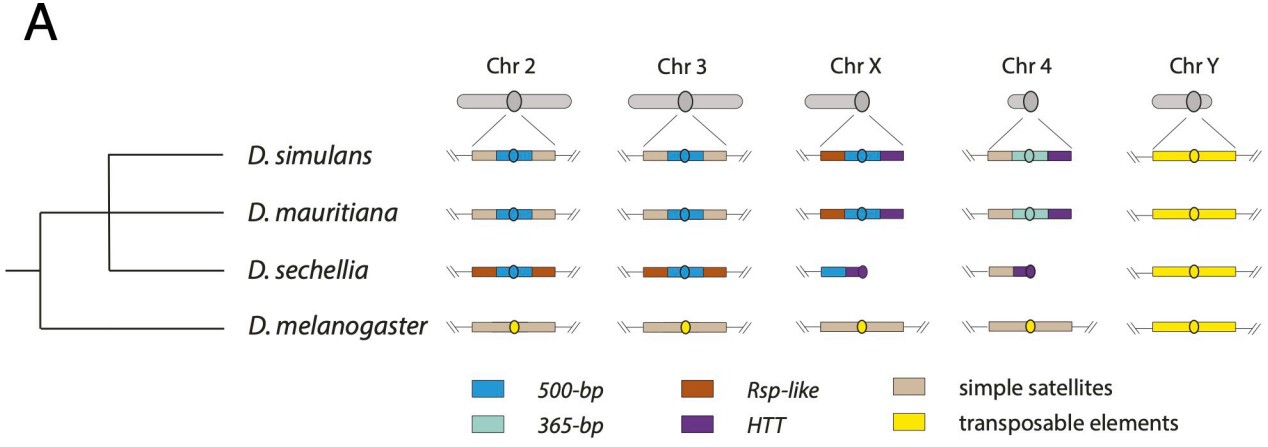

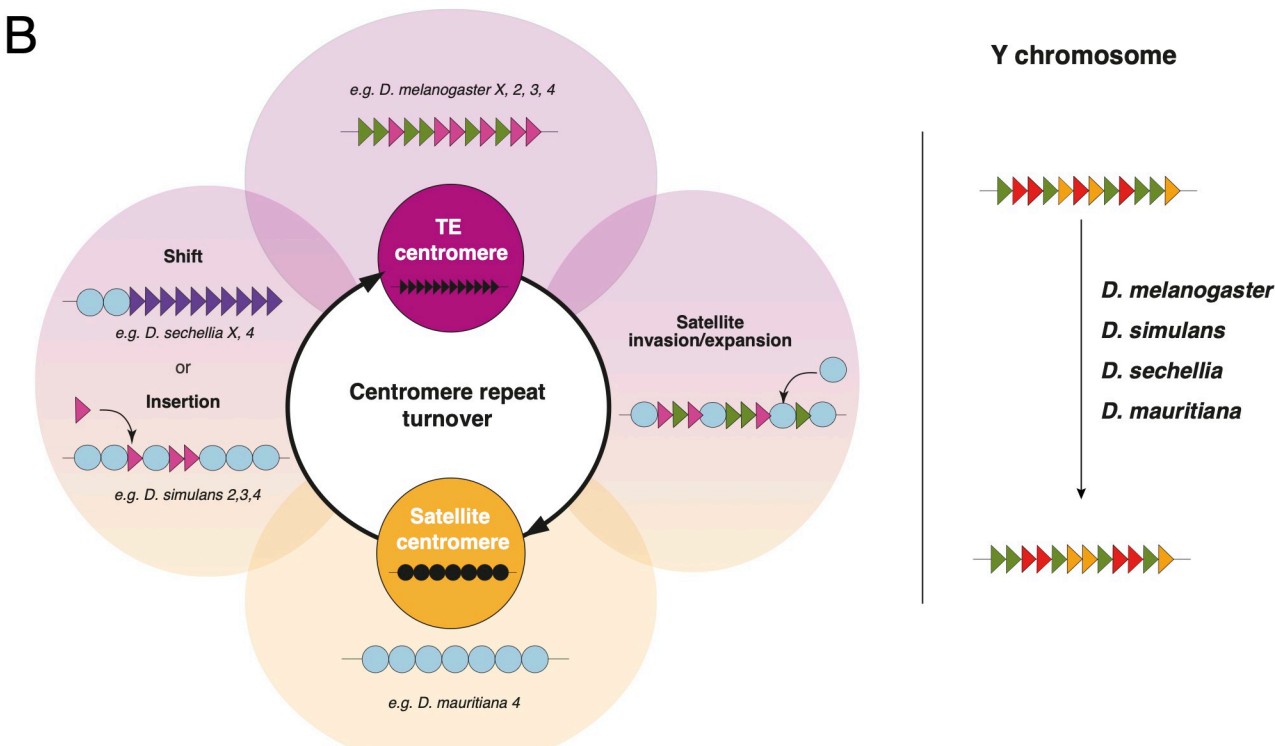

**Fig 7. Shifting centromere composition in the *D. simulans* clade species and *D. melanogaster*.** (**A**) Schematic illustration of the centromere structure and composition in the *melanogaster* clade. Each chromosome's structure is depicted in gray above each column. Below, we provide a detailed view of the centromeric and pericentromeric regions for each species. The centromere is represented as a circle. Each region is color-coded based on the dominant repeat composition, with the legend at the bottom of the figure explaining the color scheme. (**B**) An evolutionary model for the centromere sequence turnover in the *melanogaster* clade species representing the cycling between retroelement-rich and satellite-rich centromeres in the *D. melanogaster* clade species. Retroelements and satellites may be engaged in their own conflicts and thus indirectly compete to occupy centromeres. Representative examples of specific replacement events in different stages of the conflicts are depicted in the outside circles. For example, while *D. melanogaster* centromeres are rich in TEs, *D. simulans* clade centromeres are now primarily occupied by satellite DNA. The satellite-rich centromeres of *D. simulans* are still targeted by *G2/Jockey-3* retroelements and *D. sechellia*'s X and dot (fourth) chromosome centromeres shifted to the specialized telomeric *HTT* retroelements. The Y chromosome centromeres do not cycle between retroelements and satellite DNAs in the simulans clade species. Despite satellite DNAs being a major component of these Y chromosomes, their centromeres remain rich in retroelements. We speculate that this is because the dynamic turnover of centromere content is driven by female-specific selection like centromere drive in female meiosis. *HTT*, *Het-A*, *TART*, and *TAHRE*; TE, transposable element.

genome, however, insertions in the centromere might not be. There may be relaxed constraints on centromere sequence evolution, particularly if DNA sequences do not play a major role in centromere functions. Alternatively, the rapid turnover in centromeric DNA sequences could be driven by selection, either of the classic variety where selection favors divergence in DNA sequences, or from selfish processes like meiotic drive. The centromere drive hypothesis predicts an evolutionary arms race between centromere sequences and centromeric proteins and might explain how a chromosome domain with essential function can evolve so rapidly [12,17]. Support for this hypothesis was originally based on centromere sequence divergence between more distantly related species and the rapid evolution of centromeric proteins [12,17]. Our study highlights how rapid this centromere sequence evolution can be. We speculate that many of the observations we made about centromere evolution in the *D. simulans* clade are consistent with a history of genetic conflict. The *365-bp* and *500-bp* satellite DNAs are clade-specific satellites that emerged recently and spread rapidly across centromeres. Expansions of these repeats could correspond to stronger centromeres that behaved selfishly, perhaps driving in female meiosis. Repeat expansions may be accompanied by the accumulation of centromeric chromatin, thus recruiting more kinetochore proteins and biasing their segregation to the oocyte, as is the case for the minor satellite at mouse centromeres [7]. The spread of *500-bp* to what is now pericentromeres may be a signature of past expansion—CENP-A may have restricted its domain to a subset of the *500-bp* satellite array to avoid centromere asymmetry. However, whether these changes occur within a stable CENP-A chromatin domain that the *500-bp* and *365-bp* complex satellites invaded, or CENP-A relocated to new sites that contained *500-bp* and *365-bp* complex satellites remains an open question. Future experimental and evolutionary genetic studies of centromere dynamics may help distinguish between these hypotheses. Regardless of driving forces behind this turnover, the rapid reorganization of centromeric sequences over short evolutionary timescales underscores the dynamic nature of centromeres and highlights their potential as hotspots for evolutionary innovation.

The X and dot chromosomes of the *melanogaster* species are classified as acrocentric based on cytological observations of mitotic chromosomes (reviewed in [43]). Here, our epigenetic profiling and high-resolution cytology allows us to distinguish between chromosomes with independent, but nearby centromere and telomere domains (e.g., in *Mus musculus* where centromeres are positioned 10 to 1,000 kb away from the telomere [60,61]), and telocentric chromosomes where centromeres and telomeres are on adjacent sequences (e.g., *Mus Pahari* [45]) or both occupy the same repetitive array. While the centromere shift to the *HTT* could be a cause or consequence of the loss of the centromeric satellite, the presence of *500-bp* satellite adjacent to the telocentromeric domain on the X chromosome (Fig 4A–4C) suggests the latter scenario. We therefore suspect that the association of the *HTT* retroelements and the centromere is due to centromere shift rather than centromere-targeted transposition. While in *D. sechellia* X and dot chromosomes are clearly telocentric, we think that centromeres are close to the telomeres in *D. simulans* and *D. mauritiana*. Our observations raise important questions regarding the respective roles of centromeres and telomeres in chromosome biology as well as their functional association. Interestingly, in fission yeast the telomere bouquet is essential for spindle formation through telomere-centrosome contacts. However, if the telomere bouquet is disrupted, centromere-centrosome contacts can rescue the spindle defect, suggesting that centromeres and telomeres have functional similarities and interchangeable roles [62]. Similarly in mice, one of the shelterin complex proteins that is essential for telomere function (TRF1) is also required for centromere and kinetochore assembly [63]. In the case of *D. sechellia*, *HTT* elements with historical telomere-specific functions now need to also carry out and avoid interfering with centromere functions, at least at the structural level.

Although the dot and X centromeres of D. *sechellia* are unique due to their association with telomere-specialized retroelements, TEs are commonly found in the centromeres of the *simulans* clade, even when satellite DNA is the predominant repeat. *G2/Jockey-3* seems to have actively targeted centromeric regions in the ancestor of *D. melanogaster* and the *D. simulans* clade, despite their disparate underlying sequence composition. This suggests that this element may target centromeric chromatin itself rather than a specific sequence. Such centromere-chromatin targeting by retroelements may also exist in maize [64,65] and *Arabidopsis* [57,66,67]. Transformation experiments in *Arabidopsis* showed that the centromere-associated *Tal1* retroelement from *A. lyrata* is able to target *A. thaliana* centromeres [66] despite divergent (30%) centromeric satellites in these species [68].

On one hand, TEs may limit harm to their host by inserting at centromeres, far from protein-coding genes and with little opportunity for deleterious ectopic recombination [27,69,70]. They may also escape host defenses by inserting in CENP-A nucleosomes [71]. However, a high density of TEs may inactivate centromeres through heterochromatinization [26,72]. On the other hand, centromeres may tolerate TEs that contribute positively to a proper chromatin and transcription environment for centromere assembly, and in a sense therefore cooperate with the genome. Indeed, there is evidence across species that RNA is important for centromere assembly [73–77]. Centromeric copies of *G2/Jockey-3* are transcribed in *D. melanogaster* [28]; therefore, these TEs might contribute to centromere function despite having properties of an opportunistic selfish genetic element.

This apparent balance between TE-mediated conflict and cooperation could play an important role in fueling rapid centromere evolution. Klein and O'Neill [27] proposed that retroelement transcription can favor the recruitment of new insertions at neocentromeres, recruiting more CENP-A to stabilize the centromere. Recurrent insertions may also facilitate the emergence, or the spread, of satellites, which if favored by selection or selfish dynamics, can become the major component of centromeres. While there might not be direct competition between retroelements and satellites, both can coexist and cooperate to allow or even facilitate centromere function, centromeres may then cycle between retroelement-rich and satellite-rich domains through repeated bouts of retroelement invasion, followed by satellite birth and satellite expansion events (Fig 7B). The centromeres that we study here might represent different stages of this cycle.

The unique composition of Y chromosome centromeres, where we do not observe centromere turnover, may be because it is the only chromosome that never experiences female meiosis (Fig 7B). While selfish centromere drivers (e.g., driving satellites) cannot invade Y chromosomes, these chromosomes still offer a safe haven for transposable element insertions. However, Y chromosomes are subject to different evolutionary pressures and mutation patterns that might affect its sequence evolution [33], although not exclusively at the centromere. Distinguishing between drive and any alternative hypotheses will require future empirical studies of chromosome transmission and the development of formal population genetic models for centromere drive.

In conclusion, we demonstrate the extremely rapid turnover of centromeric DNA in the *D. melanogaster* subgroup, which could be driven by multidimensional selfish behaviors. First, TEs can insert centromeres to ensure their own transmission without hampering host fitness. In turn, the changes in centromeric sequences could alter centromeric chromatin, and possibly bias chromosome transmission through female meiosis, e.g., centromere drive. Lastly, the high mutation rates at centromeres might further promote the birth and turnover of centromeric satellites. If the genetic elements occupying centromeres are indeed selfish, competition for centromere invasion and potential for biased transmission to the next generation can drive rapid turnover of centromere composition. In these species, retroelements and satellite DNA

may be competing, perhaps indirectly, for centromere occupancy. These dynamics have implications not just for the role of centromeric DNAs in chromosome segregation, but also for the role of retroelements in genome function, and karyotype evolution [78] broadly.

## Materials and methods

### Fly strains

For *D. sechellia* and *D. mauritiana*, we used the same sequenced strains used to build the heterochromatin enriched genome assemblies [30]: Rob12 (Cornell SKU: 14021–0248.25) and w12 (Cornell SKU: 14021–0241.151), respectively. For *D. simulans*, we used the wXD1 strain that is maintained in the Larracuente lab. While it is the same strain as the one used to build the heterochromatin enriched assembly, our isolate appears to have a structural polymorphism on the X chromosome pericentromeric compared to the assembly [33]. All the experiments conducted in this study were performed using the same isolate. For *D. melanogaster*, we used an inbred strain from the Netherlands (N25) [79].

### Dryad DOI

https://doi.org/10.5061/dryad.1zcrjdg2g [40].

### Antibodies used

The list of primary and secondary antibodies that we used for this study is details below:

Anti-CENP-A antibody (***α***-CID20): polyclonal rabbit antibody synthesized for this study (by Covance). The CENP-A antibody was raised against the MPRHSRAKRAPRPSAC peptide [8]. The final serum was proteinA purified. We used this antibody at 1:50 dilution for the CUT&Tag. We validated the specificity of the antibody by western blot (S11 Fig).

Anti-CENP-C antibody (***α***-CENP-C12): polyclonal rabbit antibody synthesis for this study (by Genscript). The CENP-C antibody was raised against the NNRRSMRRSGNVPGC peptide. The final serum was affinity purified. We used this antibody at 1:100 dilution for the Immunostaining on mitotic chromosomes.

Anti-CENP-A antibody (***α***-CIDH32): polyclonal chicken antibody, gift from the Mellone lab. We used the antibody at 1:100 dilution for the Immunostaining on chromatin fibers.

Anti-Mouse IgG H&L antibody (abcam, ab46540): rabbit antibody that we used as a negative control for the CUT&Tag at 1:100 dilution.

Anti-H3K9me3 antibody (abcam, ab176916): rabbit monoclonal antibody. We used this antibody as a positive control for the CUT&Tag at 1:100 dilution.

Anti CENP-C primary antibody: Guinea Pig antibody from [80]. We used this antibody for larval brain squashes for *G2/Jockey-3* IF-FISH at 1:500 dilution.

Guinea Pig anti-rabbit unconjugated (Novus Biologicals, NBP1-72763). We used this secondary antibody for the CUT&Tag at 1:100 dilution.

Goat anti-rabbit IgI H&L conjugate with Alexa Fluor 488 (abcam, ab150077). We used this secondary antibody for the Immunostaining on mitotic chromosomes spread at 1:500 dilution.

Goat anti-Chicken IgY (H+L) Secondary Antibody, Alexa Fluor 488 (Invitrogen, A-11039).

Goat anti Guinea Pig conjugate with AlexaFlour 546 (Thermo Catalog # A-11074). We used this secondary antibody for the Immunostaining on mitotic chromosomes spread at 1:500, for *G2/Jockey-3* IF-FISH.

## Western blot

Twenty flies from each species were homogenized in 200 μl 1× Laemmli buffer (diluted from BioRad 4× Laemmli Sample Buffer [1610747] with 2-mercaptoethanol [Sigma] and 1× Pierce EDTA-free Protease inhibitors [Thermo Fisher A32965]), denatured by incubation at 95°C for 10 min, centrifuged at 15,000 reads per million (RPM) for 5 min at 4°C, and 20 μl of each the supernatant and PageRuler Prestained Protein Ladder (Thermo Fisher [26616]) was run 4% to 15% Mini-Protean TGX gel. The protein was transferred to PVDF membrane (Novex Invitrolon [LC2005]), blocked (Li-Cor Intercept Blocking buffer [927–60001]), incubated with 1:1,000 Rabbit anti-CENP-A(lab stock), washed 3 times with TBS/0.1% Tween-20, incubated with 1:20,000 Goat Anti-Rabbit IgG (H+L) DyLight800 (Invitrogen SA5-10036), washed 3 times with TBS/0.1% Tween-20, and imaged with Li-Cor Odyssey CLx imaging system.

## CUT&Tag

**Nuclei isolation.** We collected *Drosophila* embryos overnight at 25°C in cages containing a grape juice-agar plate with yeast paste. We used 0 to 16 h embryos to perform nuclei isolation as in [81]. We washed embryos in the embryo wash buffer (0.7% NaCl, 0.04% Triton-X100) then dechorionated using 50% bleach for 30 s. We ground embryos in 1 ml buffer B (pH 7.5, 15 mM Tris-HCl, 15 mM NaCl, 60 mM KCl, 0.34 M Sucrose, 0.5 mM Spermidine, 0.1% β-mercaptoethanol, 0.25 mM PMSF, 2 mM EDTA, 0.5 mM EGTA) using a homogenizer and filtered to remove large debris. We centrifuged nuclei at 5,000g for 5 min and resuspended in 500 μl of buffer A (pH 7.5, 15 mM Tris-HCl, 15 mM NaCl, 60 mM KCl, 0.34 M Sucrose, 0.5 mM Spermidine, 0.1% β-mercaptoethanol, 0.25 mM PMSF), twice. We resuspended the final pellet in CUT&Tag wash buffer (20 mM HEPES (pH 7.5), 150 mM NaCl, 0.5 mM Spermidine) to a final concentration of 1,000,000 nuclei/ml.

**CUT&Tag.** We performed CUT&Tag using around 100,000 nuclei per sample. We used the pA-Tn5 enzyme from Epicypher and followed the manufacturer's protocol (CUT&Tag Protocol v1.5). For each species, we performed 3 replicates with the anti-CID20 antibody (1:50), one positive control using anti-H3K9me3 (1:100), and one negative control using the anti-IgG antibody (1:100).

While a spike in control would allow us to measure quantitative variation between samples, our analysis of centromere chromatin is qualitative. We therefore elected to exclude a spike in to maximize our centromere-associated read recovery.

**Library preparation.** For the library preparation, we used the primers from [82] (S8 Table). We analyzed each library on Bioanalyzer for quality control, representative profiles of CENP-A and H3K27me3 profiles are provided in S11B Fig. Before final sequencing, we pooled 2 μl of each library and performed a MiSeq run. We used the number of resulting reads from each library to estimate the relative concentration of each library and ensure an equal representation of each library in the final pool for sequencing. We sequenced the libraries in 150-bp paired-end mode on HiSeq Illumina. We obtained around 10 million reads per library, except for the IgG negative control, which usually has a lower representation (S9 Table).

## Centromere identification

We trimmed paired-end reads using trimgalore (v0.4.4) [83] (*trim_galore—paired—nextera—length 75—phred33—no_report_file–fastqc*) and assessed read quality with FASTQC.

We mapped reads against the reference genome with bwa (v7.4) using the *BWA-MEM* algorithm (default parameters). We used the heterochromatin-enriched assemblies of *D. melanogaster* [40], *D. simulans*, *D. sechellia*, and *D. mauritiana* [33]. We converted the resulting sam alignment files into bam files and sorted using respectively samtools (v1.11) *view* and *sort* command. We removed PCR duplicates using *Markduplicates* from Picardtools (v2.12.0) (https://broadinstitute.github.io/picard/). Because we are working with highly repetitive sequences, we analyzed both the unique and multi-mapping reads. We thus performed 2 different filtering based on mapping quality using samtools *view* [84]. To include multi-mapping reads, we use the following parameters: *-b -h -f 3 -F 4 -F 8 -F 256 -F 2048*. To keep only the uniquely mapping reads, we use the following parameters: *-b -h -f 3 -F 4 -F 8 -F 256 -F 2048 -q30*.

We estimated read coverage using the *bamCoverage* command from deeptools (v3.5.1) using the option—*scaleFactor -bs 1*—*extendReads* and normalized the read coverage to RPM.

We called peaks based on fragment size using MACS2 callpeak [85] (v2017-10-26) (option -f BAMPE -g dm -q 0.01 -B—call-summits) and performed an IDR analysis (https://github.com/nboley/idr) to identify high confidence peaks that overlapped between replicates (IDR <0.05, S1 Table). The localization of these high confident peaks allowed us to identify the candidate centromere contigs (S1 Fig).

We calculated mappability along each centromere candidate contig using GenMap (https://github.com/cpockrandt/genmap) with 150-mers to mimic read length.

## Repeat enrichment analyses

For this analysis, we used the multi-mapping bam file. We annotated the reference genome (S1–S4 Files) using a custom repeat library specific to each species (S5–S8 Files) with Repeatmasker [86] (options *-no_is -a -inv -pa 20 -div 20*). Using htseq-count [87], we counted the number of reads that map to each repeat and calculated RPM. To determine the enrichment, we normalized the RPM counts for CENP-A by RPM counts for IgG (negative control). The 25% most enriched repeats are presented in S10 Table, and the top 20 most enriched repeats among all replicates are presented in Fig 1B, 1E, and 1H.

To explore origins of the centromeric complex satellites we blasted (*blastn* with default parameter) the consensus sequences of *500-bp*, *136-bp*, and *365-bp* satellites against the genome of *D. melanogaster* [47], the *simulans* clade [33], and more distant species, *D. yakuba*, *D. annanassae*, *D. pseudoobscura*, *D. erecta*, and *D. virilis* [88]. All hits are reported in S3–S5 Tables.

The dotplots of the Y chromosome centromeres cenY (S9 Fig) were generated using re-DOT-able v1.1 (https://www.bioinformatics.babraham.ac.uk/projects/redotable/).

## *G2/Jockey-3* evolutionary analyses

We surveyed *G2/Jockey-3* evolution in additional species with improved genome assemblies of *D. simulans*, *D. sechellia*, and *D. mauritania* [89] and publicly available Nanopore assemblies of *D. yakuba*, *D. erecta*, and *D. ananassae* [90]. We identified *G2/Jockey-3* sequences with 2 complementary methods. First, we annotated each genome assembly with our custom *Drosophila* TE library including the *D. melanogaster G2/Jockey-3* consensus sequence [71] using Repeatmasker v4.1.0. The annotations and 500-bp flanking regions were extracted with BEDTools v2.29.0 [81] and aligned with MAFFT [91] to generate a species-specific consensus sequence with Geneious v.8.1.6 [92]. Each assembly was annotated again using Repeatmasker with the appropriate species-specific *G2/Jockey-3* consensus sequence. Second, we constructed de novo repeat libraries for each species with RepeatModeler2 v.2.0.1 [93] and identified candidate *G2/Jockey-3* sequences which shared high similarity with *G2/Jockey-3* in *D.*

*melanogaster* identified with BLAST v.2.10.0. We did the same with *Jockey-1* (LINEJ1_DM) as confirmation of our methods and to use it as an outgroup for the TE fragment alignment. We removed candidates shorter than 100 bp from the analysis. We identified ORFs within consensus TE sequences with NCBI ORFfinder. We used Repeatmasker to annotate the genome assemblies with the de novo *Jockey-3* consensus sequences. To infer a phylogenetic tree of TEs, we aligned *G2/Jockey-3* fragments identified in each species with MAFFT and retained sequences corresponding to the ORF bounds of the consensus sequences. We removed ORF fragments <400 bp. We inferred the tree with RAxML v.8.2.11 [94] using the command "raxmlHPC-PTHREADS -s alignment_Jockey-3_melsimyak_400_ORF2_mafft.fasta -m GTRGAMMA -T 24 -d -p 12345 -# autoMRE -k -x 12345 -f a."

## Oligopaint design and synthesis

We designed Oligopaint probes targeting *500-bp*, *136-bp*, *365-bp*, *Rsp-like*, *HTTs*, and the Y centromere islands of each species using ProbeDealer [95] with some modifications. We extracted the fasta sequences containing the target repeat from the reference genomes and used it as the input for ProbeDealer. After designing all the possible oligo probes, ProbeDealer usually maps them back against the reference genome to eliminate multi-mapping oligos. Because we are working with highly repetitive sequences, we skipped this step. We mapped the oligos to the reference genome to manually inspect for potential off targets. The final oligo list is in S11 Table. Oligopaints libraries were synthesized by Genscript. We then synthesized and purified each Oligopaint sublibrary as described in [29].

## IF-FISH on mitotic chromosome

We dissected brains from third instar larvae (both sexes) in PBS, incubated 8 min in 0.5% sodium citrate. We fixed for 6 min in 4% formaldehyde, 45% acetic acid before squashing. We squashed the brains between a poly-lysine slide and coverslip and before immersing in liquid nitrogen. After 5 min in PBS and 10 min in PBS, we blocked slides for at least 30 min in blocking buffer (3%BSA, 1% goat serum in PBST). For immunofluorescence (IF), we incubated slides in primary antibody ($\alpha$-CENP-C12 1:100) overnight at 4°C. We washed slides 3 times for 5 min in PBST. We incubated slides in secondary antibody (anti-rabbit 1:500) for 1 to 3 h at room temperature and washed 3 times for 5 min in PBST. We post-fixed the slides using 10% formaldehyde diluted in 4XSSC, incubating 20 min at room temperature and washed 3 times for 3 min with 4XSSC and one time for 5 min with 2XSSC. For the hybridization, we used 20 pmol of primary probes (S11 Table) and 80 pmol of the secondary probes (S12 Table) in 50 µl of hybridization buffer (50% formamide, 10% dextran sulfate, 2XSSC). We heated slides for 5 min at 95°C to denature and incubated them overnight at 37°C in a humid chamber. We then washed the slides 3 times for 5 min with 4XSSCT and 3 times for 5 min with 0.1SSC before mounting in slowfade DAPI.

We use acetic acid to obtain high-quality chromosome spreads; however, this also removes histones. Thus, it is not feasible to perform IF on mitotic spread using anti-histone antibodies, such as CENP-A. We therefore use CENP-C—a kinetochore protein that marks centromeres and overlaps with CENP-A [37].

## IF-FISH on chromatin fibers

We dissected third instar larval brains in 1XPBS (3 to 4 brains per slide) and incubated in 250 µl of 0.5% sodium citrate with 40 µg of dispase-collagenase, for 12 min at 37°C. The tissue was transferred to a poly-lysine slide using Shandon Cytospin 4 at 1,200 RPM for 5 min. We positioned slides vertically in a tube containing the Lysis buffer (500 nM NaCl, 25 mM Tris-

HCL (pH 7.5), 250 nM Urea, 1% Triton X-100) and incubated for 16 min. For the fiber stretching, we allow the buffer to slowly drain from the tube with the hole at the bottom (by removing the tape). A steady flow rate will generate a hydrodynamic drag force which generates longer and straighter fibers. We incubated slides in a fixative buffer (4% formaldehyde) for 10 min and then 10 min in 1XPBST (0.1% Triton). For the IF, we first blocked the slides for 30 min in blocking buffer (1.5% BSA in 1XPBS). We incubated slides overnight at 4˚C with the primary antibody (*α*-CIDH32, 1:100) and washed 3 times for 5 min in 1xPBST. We incubated slides with the secondary antibody (anti-chicken, 1:500) for 1 to 3 h at room temperature and washed 3 times for 5 min with 1XPBST. We post-fixed the slide with 10% formaldehyde for 20 min and washed 3 times for 5 min in 1XPBST. We then incubated slides for 10 min in 2XSSCT at room temperature and 10 min in 2XSSCT—50% formamide at 60˚C. For the hybridization, we used 40 pmol of primary probes (S11 Table) and 160 pmol of the secondary probes (S12 Table) in 100 µl of hybridization buffer (50% formamide, 10% dextran sulfate, 2XSSC). We heated slides for 5 min at 95˚C to denature and incubated them overnight at 37˚C in a humid chamber. We then washed the slides 15 min with 2XSSCT at 60˚C, 15 min with 2XSSCT at room temperature, and 10 min with 0.1XSSC at room temperature. We incubated slides for 5 min in DAPI (1 mg/ml) before mounting in SlowFade Gold (Invitrogen S36936).

### *G2/Jockey-3* IF-FISH

*D. simulans*, *D. sechellia*, and *D. mauritiana* third instar larval brains were dissected in 1× PBS and all attached tissue or mouth parts were removed with forceps. Brains were immersed in 0.5% sodium citrate solution for 8 min in a spot well dish. The tissue was placed in a 6 µl drop of 45% acetic acid, 2% formaldehyde on a siliconized (Rain X) coverslip for 6 min. A poly-lysine–coated slide was inverted and placed on the brains to make a sandwich. After flipping the slide and gently removing excess fixative between a bibulous paper, the brain was squashed using the thumb by firmly pressing down. Slides were then immersed in liquid nitrogen and the coverslip flipped off using a razor blade and transferred to 1× PBS for 5 min to rehydrate before proceeding with IF-FISH. Slides were then washed with 1× PBST (0.1% Triton X-100) for 5 min on a rotator, repeated 3 times. Slides were then transferred to a coplin jar containing blocking solution (1% BSA in 1× PBST) for 30 min while rocking. Diluted antibodies were applied to the slides coating the brains with 50 µl of primary antibodies, covered with parafilm and stored in a dark chamber at 4˚C overnight. The following day, slides were washed 4 times with 1× PBST for 5 min while rocking. Secondary antibodies diluted with block were applied to the brains and covered with parafilm, then incubated at room temperature for 1 h. After the 1 h incubation, slides were washed 4 times in 1× PBST for 5 min while rotating. Slides were then post-fixed with 3.7% formaldehyde diluted with 1× PBS for 10 min in the dark. Slides were washed for 5 min in 1× PBS while rotating before proceeding to FISH. The following FISH protocol for G2/Jockey-3 labeling and the synthesis of the *G2/Jockey-3* probe was performed as described in the methods of Chang and colleagues [29]. Slides were dehydrated in an ethanol row (3 min washes in 70%, 90%, and 100% ethanol) and allowed to air-dry completely for a few minutes. Probe mix (20 µl) containing 2xSSC, 50% formamide (Sigma-Aldrich), 10% dextran sulfate (Merck), 1 µl RNase cocktail (Thermo Fisher), and 100 ng of DIG-labeled *G2/Jockey-3* probe was boiled at 80˚C for 8 min, incubated on ice for 5 min, and then applied to slides, covered with a glass coverslip, and sealed with paper cement. Sealed slides were denatured on a slide thermocycler for 5 min at 95˚C and incubated at 37˚C overnight to hybridize. Slides were then washed 3 times in a coplin jar for 5 min in 2xSSC, 50% formamide at 42˚C. Slides were then washed 3 times for 5 min in 0.1xSSC at 60˚C, and then blocked in block buffer 1% BSA, 4xSSC, 0.1% Tween-20 at 37˚C for 45 min. Slides were

incubated with 50 μl of block buffer containing a fluorescein-labeled anti-DIG antibody (sheep, 1:100, Roche) for 60 min at 37˚C. Slides were then washed 3 times for 5 min in 4xSSC, 0.1% Tween-20 at 42˚C. Slides were washed with 1× PBS briefly in a coplin jar and finally mounted on a coverslip with Slowfade and DAPI, then sealed with nail polish.

### Image acquisition

We imaged using a LEICA DM5500 microscope with a 100×/oil immersion objective or Delta vision using an Olympus UPLansApo 100×/1.40 oil immersion objective, maintaining all exposures consistent across each experiment. Images obtained with the Deltavision microscope were deconvolved with Softoworks using 5 iterations with the "conservative" setting. Images were edited, cropped, and pseudocolored using Fiji.

### Supporting information

**S1 Fig. CUT&Tag results from the two additional CENP-A replicates (top 2 row) and the IgG negative control (third row) and the mappability score (bottom row) for each centromere in *D. simulans*.** The y-axis represents the normalized CENP-A or IgG enrichment in RPM. Black and gray plotted lines represent the enrichment based on uniquely mapping and all reads (including multi-mappers), respectively. The black and gray tracks below each plot correspond to MACS2 peaks showing significantly enriched regions based on the uniquely mapping and all reads (including multi-mappers), respectively. The precise locations of all peaks are listed in S1 Table. The colored cytoband at the bottom of the plot shows the repeat organization. The color code is shown in the legend at the bottom of the figure. The data underlying this figure can be found at https://doi.org/10.5061/dryad.1zcrjdg2g [40]. (PDF)

**S2 Fig. CUT&Tag results from the 2 additional CENP-A replicates (top 2 row) and the IgG negative control (third row) and the mappability score (bottom row) for each centromere in *D. sechellia*.** The y-axis represents the normalized CENP-A or IgG enrichment in RPM. Black and gray plotted lines represent the enrichment based on uniquely mapping and all reads (including multi-mappers), respectively. The black and gray tracks below each plot correspond to MACS2 peaks showing significantly enriched regions based on the uniquely mapping and all reads (including multi-mappers), respectively. The precise locations of all peaks are listed in S1 Table. The colored cytoband at the bottom of the plot shows the repeat organization. Color code is shown in the legend at the bottom of the figure. The data underlying this figure can be found at https://doi.org/10.5061/dryad.1zcrjdg2g [40]. (PDF)

**S3 Fig. CUT&Tag results from the 2 additional CENP-A replicates (top 2 row) and the IgG negative control (third row) and the mappability score (bottom row) for each centromere in *D. mauritiana*.** The y-axis represents the normalized CENP-A or IgG enrichment in RPM. Black and gray plotted lines represent the enrichment based on uniquely mapping and all reads (including multi-mappers), respectively. The black and gray tracks below each plot correspond to MACS2 peaks showing significantly enriched regions based on the uniquely mapping and all reads (including multi-mappers), respectively. The precise locations of all peaks are listed in S1 Table. The colored cytoband at the bottom of the plot shows the repeat organization. The color code is shown in the legend at the bottom of the figure. The data underlying this figure can be found at https://doi.org/10.5061/dryad.1zcrjdg2g [40]. (PDF)

**S4 Fig. Location of the peaks resulting from the IDR analysis—significantly enriched region conserved between the 3 replicates. The y axis represents the sum of the peaks length for each contig. The contig corresponding to the centromere are colored in black. The data underlying this figure can be found in S1 Table.**
(TIF)

**S5 Fig. CUT&Tag results from the 3 CENP-A replicates (top 2 row) and the IgG negative control (bottom row) for each centromere in *D. melanogaster*. The y-axis represents the normalized CENP-A or IgG enrichment in RPM. Black and gray plotted lines represent the enrichment based on uniquely mapping and all reads (including multi-mappers), respectively.** The black and gray tracks below each plot correspond to MACS2 peaks showing significantly enriched regions based on the uniquely mapping and all reads (including multi-mappers), respectively. The precise locations of all peaks are listed in S1 Table. The colored cytoband at the bottom of the plot shows the repeat organization. The color code is shown in the legend at the bottom of the figure. The data underlying this figure can be found at https://doi.org/10.5061/dryad.1zcrjdg2g [40].
(TIF)

**S6 Fig.** (A) IF-FISH on mitotic chromosomes from the larval brain with CENP-C antibody and *500bp* and *136-bp* probes. The inset represents a zoom on each centromere. (B) IF-FISH on mitotic chromosomes from the larval brain from *D. sechellia* with CENP-C antibody and *365-bp* and *AATAT* probes. The inset represents a zoom on the dot chromosome centromere.
(TIF)

**S7 Fig. Distribution of the percentage of divergence of individual insertion from the consensus sequence for each centromeric satellite. Only insertions with a length >80% of consensus length were kept. The percentage of divergence was extracted from the Blast output. The data underlying this figure can be found in S3–S5 Tables.**
(TIF)

**S8 Fig. IF-FISH on chromatin fibers from the *D. sechellia* larval brains with CENP-A antibody and *500-bp* and *HTT* probes.** A representative image of each centromere pattern is presented along with the total number of images collected for each pattern. CENP-A is present on the *HTT* region with or without *500-bp* flanking, corresponding to the X and dot chromosome, respectively. CENP-A is also present on a *500-bp* region, corresponding to the autosomal centromeres and without 500-bp nearby, consistent with the Y chromosome.
(TIF)

**S9 Fig. Dotplot from the alignment on the Y chromosome centromere against itself to highlight higher order repeat.** The Dotplot was produced using re-DOT-able with a sliding window of 100 bp. The cytoband below each dotplot represent the repeat composition of the region. The color code is indicated in the legend.
(TIF)

**S10 Fig. Phylogenetic tree with node support of consensus *G2/Jockey-3* ORF sequences in relation to closely related Jockey elements. Closely related Jockey elements were identified from [48].** Three *D. yakuba* fragments which span the >50% of the ORF are also included. The data underlying this figure can be found at https://doi.org/10.5061/dryad.1zcrjdg2g [40].
(TIF)

**S11 Fig. CENP-A antibody validation.** (A) Western blots using our custom-generated CENP-A antibody on samples from all 4 species *D. melanogaster* clade species. (B) Bioanalyzer

profile of the CUT&Tag libraries obtained for our custom-generated CENP-A and H2K27me3 antibodies.
(TIF)

**S1 Table. Output of the IDR analyses for both the uniquely (Q30) and all reads (including multi-mappers) (Q0) peak calling for each species.**
(XLSX)

**S2 Table. Coordinate of each centromere of the *melanogaster* clade.**
(XLSX)

**S3 Table. Output of the blast for *500-bp*.**
(XLSX)

**S4 Table. Output of the blast for *136-bp*.**
(XLSX)

**S5 Table. Output of the blast for *365-bp*.**
(XLSX)

**S6 Table. Quantification of chromatin fibers.**
(XLSX)

**S7 Table. Composition of the Y chromosome centromeres.**
(XLSX)

**S8 Table. List of the primers used for the CUT&Tag libraries.**
(XLSX)

**S9 Table. Number of reads and percentage of mapping for each CUT&Tag libraries.**
(XLSX)

**S10 Table. Repeat enrichment in RPM.**
(XLSX)

**S11 Table. Oligopaints sequences.**
(XLSX)

**S12 Table. Sequences of probes used for FISH.**
(XLSX)

**S1 File. *D. simulans* genome annotation.**
(GFF)

**S2 File. *D. mauritiana* genome annotation.**
(GFF)

**S3 File. *D. sechellia* genome annotation.**
(GFF)

**S4 File. *D. melanogaster* genome annotation.**
(GFF)

**S5 File. *D. simulans* repeat library.**
(FASTA)

**S6 File. *D. mauritiana* repeat library.**
(FASTA)

**S7 File. *D. sechellia* repeat library.**
(FASTA)

**S8 File. *D. melanogaster* repeat library.**
(FASTA)

## Acknowledgments

We would like to thank the members of the Larracuente and Mellone labs for helpful discussion and Emiliano Martí for comments on the manuscript. We are grateful to the University of Rochester Center for Integrated Research Computing for access to computing cluster resources and the University of Rochester Genomics Research Center for sequencing services.

## Author Contributions

**Conceptualization:** Cécile Courret, Ching-Ho Chang, Barbara G. Mellone, Amanda M. Larracuente.

**Data curation:** Cécile Courret, Xiaolu Wei, Xuewen Geng.

**Formal analysis:** Cécile Courret, Lucas W. Hemmer, Amanda M. Larracuente.

**Funding acquisition:** Amanda M. Larracuente.

**Investigation:** Cécile Courret, Ching-Ho Chang, Barbara G. Mellone, Amanda M. Larracuente.

**Methodology:** Cécile Courret.

**Project administration:** Amanda M. Larracuente.

**Resources:** Barbara G. Mellone, Amanda M. Larracuente.

**Software:** Cécile Courret.

**Supervision:** Cécile Courret, Barbara G. Mellone, Amanda M. Larracuente.

**Validation:** Cécile Courret, Lucas W. Hemmer, Xiaolu Wei, Prachi D. Patel, Nicholas J. Fuda.

**Visualization:** Cécile Courret, Prachi D. Patel, Bryce J. Chabot.

**Writing – original draft:** Cécile Courret, Barbara G. Mellone, Amanda M. Larracuente.

**Writing – review & editing:** Cécile Courret, Ching-Ho Chang, Barbara G. Mellone, Amanda M. Larracuente.

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
