## [Editor Report · Decision Letter 0]

23 Aug 2024

Dear Dr Larracuente, 

Thank you for submitting your manuscript entitled "Rapid turnover of centromeric DNA reveals signatures of genetic conflict in Drosophila" for consideration as a Research Article by PLOS Biology.

Your revisions have now been evaluated by the PLOS Biology editorial staff, and I'm writing to let you know that we would like to send your submission out for re-review.

However, before we can send your manuscript back to reviewers, we need you to complete your submission by providing the metadata that is required for full assessment. To this end, please login to Editorial Manager where you will find the paper in the 'Submissions Needing Revisions' folder on your homepage. Please click 'Revise Submission' from the Action Links and complete all additional questions in the submission questionnaire.

Once your full submission is complete, your paper will undergo a series of checks in preparation for re-review. After your manuscript has passed the checks it will be sent out for review. To provide the metadata for your submission, please Login to Editorial Manager (https://www.editorialmanager.com/pbiology) within two working days, i.e. by Aug 27 2024 11:59PM.

Kind regards,

Roli Roberts

Roland Roberts, PhD

Senior Editor

PLOS Biology

rroberts@plos.org

---

## [Decision Letter · Decision Letter 1]

25 Sep 2024

Dear Dr Larracuente,

Thank you for your patience while we considered your revised manuscript "Rapid turnover of centromeric DNA reveals signatures of genetic conflict in Drosophila" for publication as a Research Article at PLOS Biology. This revised version of your manuscript has been evaluated by the PLOS Biology editors, the Academic Editor, and two of the original reviewers.

You'll see that reviewer #1 continues to be positive, but still has two requests. One is to do a formal analysis of between-replicate similarity. The other is to incorporate “mappability” into the Figs, either by normalising by a mappability index, or by including a mappability track for visual comparison. Reviewer #2 thinks that the paper is much improved, but still says that you can’t support the “reveals signatures of genetic conflict” aspect of the Title; instead s/he thinks that any mechanistic claims should be moved to the Discussion and clearly flagged as speculative. S/he also has two more minor requests; one is to better contextualise your CUT&Tag results in the Figs, and the other is to drop the inconsistent fiber-IF/FISH from Fig 4D.

Based on the reviews, we are likely to accept this manuscript for publication, provided you satisfactorily address the remaining points raised by the reviewers. Please also make sure to address the following data and other policy-related requests.

IMPORTANT - please attend to the following:

a) After discussion with the Academic Editor, we agree with reviewer #2's suggestion that you remove the "conflict" claim from the Title. To make the Title more explicit and appealing to our broader readership, we suggest that you change it to something like "Turnover of retroelements and satellite DNAs results in profound centromere reorganization over short evolutionary timescales in Drosophila species"

b) Please address the remaining requests from the reviewers. In addition, the Academic Editor made the following request about your Discussion: "For the discussion, I suggest that the authors begin with the two major hypotheses to explain rapid evolution 1) lack of purifying selection or 2) positive selection (including that which arises from genetic conflict). If the discussion is framed around these alternatives it will give the authors the opportunity to more clearly lay out what aspects of their data they think relate to genetic conflict as well as other aspects that may reflect lack of purifying selection. For example, the total lack of conservation of centromere sequences must in part reflect an absence of purifying selection on the sequence itself. I think this frame work will also reduce any appearance of bias on the part of the authors towards a conflict hypothesis, while still allowing plenty of space for thoughtful reflection on their results."

c) Please address my Data Policy requests below; specifically, we need you to supply the numerical values underlying Figs 1ABDEGH, 2AB, 3AB, 4AB, 5ABC, 6A (treefile), S1, S2, S3, S4, S5, S7, S9, S10 (treefile), S11B, either as a supplementary data file or as a permanent DOI’d deposition. I note that you have plans for a Dryad deposition, but we will need to check this. Please could you complete this deposition with the data and code needed to recreate the Figure?

d) Please cite the location of the data clearly in all relevant main and supplementary Figure legends, e.g. “The data underlying this Figure can be found in S1 Data” or “The data underlying this Figure can be found in https://zenodo.org/records/XXXXXXXX

e) Please make any custom code available, either as a supplementary file or as part of your data deposition. You already mention a Github deposition, but this is not currently accessible (we will need to check it). Also, because Github depositions can be readily changed or deleted, please make a permanent DOI’d copy (e.g. in Zenodo, or as part of your planned Dryad deposition) and provide this URL (see below).

We expect to receive your revised manuscript within two weeks. 

*Published Peer Review History*

*Press*

Sincerely,

Roli Roberts

Roland Roberts, PhD

Senior Editor

rroberts@plos.org

PLOS Biology

DATA POLICY:

Regardless of the method selected, please ensure that you provide the individual numerical values that underlie the summary data displayed in the following figure panels as they are essential for readers to assess your analysis and to reproduce it: Figs 1ABDEGH, 2AB, 3AB, 4AB, 5ABC, 6A (treefile), S1, S2, S3, S4, S5, S7, S9, S10 (treefile), S11B. NOTE: the numerical data provided should include all replicates AND the way in which the plotted mean and errors were derived (it should not present only the mean/average values).

CODE POLICY

We require the original, uncropped and minimally adjusted images supporting all blot and gel results reported in an article's figures or Supporting Information files. We will require these files before a manuscript can be accepted so please prepare and upload them now. Please carefully read our guidelines for how to prepare and upload this data: https://journals.plos.org/plosbiology/s/figures#loc-blot-and-gel-reporting-requirements

DATA NOT SHOWN?

REVIEWERS' COMMENTS:

Reviewer #1:

The revised manuscript by Courret et al. addresses many of my prior concerns, and I thank the authors for their thoughtful and detailed responses to my earlier critiques. However, I still have two lingering comments about the CENP-A CUT&Tag analysis: 

First, while the authors generate replicate CENP-A CUT&Tag libraries for each species, there is no formal analysis of similarity across replicate samples. Visual similarity across replicates is apparent, but this could simply reflect variation in mappabilty across the centromeric locus (see my second major point below). I think the paper would benefit from a more formal analysis showing similarity across replicate libraries. For instance, a PC analysis of CUT&Tag sequences could help demonstrate that CENP-A CUT&Tag findings are reproducible and robust to technical artifacts. 

Second, the CENP-A association landscape plots in Figs 1-5 are difficult to interpret. The rationale for independently surveying both uniquely mapping reads and all reads mapping to these regions is logical and sound, but the resulting CENP-A association and peak profiles are likely largely dictated by the "mappability" of these genomic regions (i.e., the density of sequences within the reference where reads can be uniquely assigned). For example, in regions where CENP-A associates with centromeric chromatin but reads cannot be uniquely mapped, peaks will be inferred in the multimapping, but not the uniquely mapping, data. Conversely, peaks inferred from pileups of uniquely mapping reads may simply represent regions where reads can be placed uniquely, rather than bona fide sites of CENP-A enrichment. In view of these limitations, CENP-A peak calling ought to be standardized by a "mappability" criterion, in addition to the number of sequenced reads. Alternatively, it could be helpful to include a track that highlights centromere regions where reads can be uniquely mapped. 

Despite these criticisms, the author's overall robust approach, which relies on the summation of evidence across multiple bioinformatic methods and cytology experiments, provides assurance that major conclusions are not artifacts of biases or limitations of one analysis. For this reason, I am overall enthusiastic about this work. I also think the authors' focused edits to the manuscript help to more strongly emphasize the novel evolutionary trends they uncover in centromere architecture. 

In addition to these points, I have a few minor comments: 

Page 3, Lines 64-65 - "The overexpression of CENP-A…": This sentence stands out as a non-sequitur to me and could (optionally) be removed. 

Page 4, Lines 90-92: This is true for some, but not all, mouse centromeres. See, e.g., Chmatal et al. 2014 Curr Biology (PMID: 25242031). 

Page 5, Lines 125-127: "These species diverged…" It may be helpful to present divergence times in generations, rather than years, to frame the extent of divergence across the surveyed Drosopholids on a common scale with other taxa. This suggestion also carries over to the comparative discussion presented on lines 493-498. Surely 2.4MY of Drosophila divergence equates to far more generations than represented across 7 MY of great ape divergence, possibly accounting for the relative conservation of centromere satellite sequence architecture in the latter!

Page 12, line 348 - Typo: space between "100" and "%" can be deleted. 

Line 516 - Typo: Mus musculus

Lines 613-614 - "Lastly, the high duplication rates of centromere…" Here, the word centromere should be plural. I also find the statement that centromeres have "high duplication rates" a bit exclusionary, as centromeres also have high deletion and single nucleotide mutation rates. Perhaps it would be more precise to modify this sentence to "Lastly, the high mutation rates of centromeres…" 

Figure 1. It is not apparent what the two plots in panels A, D, and G represent. In all cases, both plots are labeled "Autosome 2/3" and show distinct repeat composition and peak positions. 

Figure 7. The blue color assigned to the 500bp satellite sequence in the figure legend does not correspond to the blue in the diagram. 

Reviewer #2:

The revised manuscript by Larracuente and colleagues addressed many of the presentation and text related issues raised in the first round of revision. It is much clearer now what experiments were done. The way experiments are presented in the main text is also much improved, and the result section generally matches what the experiments test and the data show. (See several remaining issues below.) In its current form, I believe the datasets produced by the authors will be of high interest to researchers interested in centromere and chromosome evolution. 

However, the data does not allow the authors to make many of the mechanistic claims in the paper, and indeed these claims are not directly tested against clear null hypotheses. For instance, the current title of the manuscript is not supported by experiments. A more precise title would be "Rapid turnover of centromeric DNA in Drosophila". Throughout, claims regarding any oogenesis-specific effect (e.g., genetic conflict) should be moved to the Discussion and presented in a way that clarifies their highly speculative nature. Similarly, I am not convinced that the work sheds significant light on how changes to DNA sequence versus protein sequence regulate centromere function & evolution, or as to the potentially different centromeric functions of TEs versus satellites. 

To reiterate, I believe the dataset produced by the authors represent a significant addition to what is known about Drosophila centromeres. My concern is with regards to the mechanistic claims on centromere evolution. Whether a manuscript stripped off these mechanistic speculation merits publication in PLoS Biology is a question I leave for the editors. 

Other points: 

- The plots representing the Cenp-A CUT&Tag results (e.g., Fig. 1A,D,G) are still hard to interpret without information that places them in the context of the complete chromosomes. The x-axis in these plots refers to 'position' but lack any context. That would be confusing at best, and probably uninterpretable, especially in a wide-readership journal. This is particularly confusing given that the text also refers to other size ranges (e.g., the paragraph in lines 181-190). The data presentation should make it clear how do the 100-500kb regions that are shown relate to the 60Mb regions described in the text. 

- The explanation given for the inconsistent data in Fig. 4D (the fiber-IF/FISH) is unsatisfying. If this isn't a representative image, then why is it shown? And if this is the result of a "technical artifact" (lines 338-343), then why are the authors choosing to trust other aspects of the data? If this piece of data is crucial to the authors' story, this needs to be clarified, and analysis of the robustness of this method should be presented. If it is no longer a key part of the story, given that the authors modified their claim of 'truly telocentric' centromeres, then this piece of data should be removed from the current manuscript. Once verified, it could be published together with the results of the ongoing experiments that directly address this hypothesis.

---

## [Editor Report · Decision Letter 2]

22 Oct 2024

Dear Dr Larracuente,

Thank you for the submission of your revised Research Article "Turnover of retroelements and satellite DNA drives centromere reorganization over short evolutionary timescales in Drosophila" for publication in PLOS Biology. On behalf of my colleagues and the Academic Editor, Erin Kelleher, I'm pleased to say that we can in principle accept your manuscript for publication, provided you address any remaining formatting and reporting issues. These will be detailed in an email you should receive within 2-3 business days from our colleagues in the journal operations team; no action is required from you until then. Please note that we will not be able to formally accept your manuscript and schedule it for publication until you have completed any requested changes.

Sincerely,

Roli Roberts

Senior Editor

PLOS Biology

rroberts@plos.org